# Pancreatic cancers suppress negative feedback of glucose transport to reprogram chromatin for metastasis

Matthew E. Bechard[1], Rana Smalling [1], Akimasa Hayashi [2], Yi Zhong [2], Anna E. Word[1], Sydney L. Campbell[3], Amanda V. Tran[1], Vivian L. Weiss [1], Christine Iacobuzio-Donahue [2], Kathryn E. Wellen [3] & Oliver G. McDonald [1,4✉]

Although metastasis is the most common cause of cancer deaths, metastasis-intrinsic dependencies remain largely uncharacterized. We previously reported that metastatic pancreatic cancers were dependent on the glucose-metabolizing enzyme phosphogluconate dehydrogenase (PGD). Surprisingly, PGD catalysis was constitutively elevated without activating mutations, suggesting a non-genetic basis for enhanced activity. Here we report a metabolic adaptation that stably activates PGD to reprogram metastatic chromatin. High PGD catalysis prevents transcriptional up-regulation of thioredoxin-interacting protein (TXNIP), a gene that negatively regulates glucose import. This allows glucose consumption rates to rise in support of PGD, while simultaneously facilitating epigenetic reprogramming through a glucose-fueled histone hyperacetylation pathway. Restoring TXNIP normalizes glucose consumption, lowers PGD catalysis, reverses hyperacetylation, represses malignant transcripts, and impairs metastatic tumorigenesis. We propose that PGD-driven suppression of TXNIP allows pancreatic cancers to avidly consume glucose. This renders PGD constitutively activated and enables metaboloepigenetic selection of additional traits that increase fitness along glucose-replete metastatic routes.

[1] Department of Pathology, Microbiology, and Immunology, Vanderbilt University Medical Center, Nashville, TN, USA. [2] David M. Rubenstein Center for Pancreatic Cancer Research, Memorial Sloan Kettering Cancer Center, New York, NY, USA. [3] Department of Cancer Biology, Abramson Cancer Family Institute, University of Pennsylvania Perelman School of Medicine, Philadelphia, PA, USA. [4] Epithelial Biology Center; Vanderbilt-Ingram Cancer Center, Vanderbilt University Medical Center, Nashville, TN, USA. ✉email: oliver.g.mcdonald@vumc.org

Metastasis remains one of the least understood aspects of cancer biology[1,2]. Genetic drivers are often shared between metastases and the primary tumor that seeds them in untreated patients[3–5], raising questions as to whether metastasis-intrinsic dependencies are selected[6,7]. This is particularly relevant for distant metastasis, a multistep cascade that requires cells to escape the confines of the primary tumor, disseminate in the circulation, seed foreign soils of other organs, and achieve successful metastatic outgrowth at those sites[1,2,6,7]. A striking example is pancreatic ductal adenocarcinoma (PDAC)[8]. In PDAC patients, the primary tumor grows silently as a solitary mass in the pancreas over a period of years[5], during which time tumor cells continually disseminate[9]. Nevertheless, clinically relevant distant metastases present suddenly and are seeded by the latest evolving subclones in the primary tumor[5,10]. Metastatic outgrowth then progresses rapidly, culminating in hundreds to thousands of metastatic tumors that diffusely involve the liver and lungs[5,11,12]. This implies genetic selection for traits that accelerate disease progression. However, proliferation rates are modest[5], and all known driver mutations[4,5,10,13–17] and consequential copy number changes[10,15,17–20] are shared between primary tumor and metastases[4,5,10], including those required for metastasis itself[21–24].

PDAC genetic drivers are selected in part to encode metabolic adaptations that support tumor growth within the densely fibrotic and nutrient-poor primary tumor stroma[25,26]. In clinical settings we noticed that similar dense fibrosis was retained in metastatic peritoneal deposits (Fig. 1a), which are seeded directly off the primary onto surrounding organ surfaces. In contrast, delicate fibrosis was present in distant metastases (Fig. 1a), which are seeded along nutrient-replete hematogenous routes after breaching dense stromal barriers[27]. The context of these clinical observations prompted us to ask if a nutrient-dependent metabolic adaptation that was not genetically encoded might support distant metastasis. We hypothesized selection for PGD activation, since distant metastases isolated from PDAC patients are strongly PGD-dependent even though PGD itself is not mutated, amplified, or recurrently overexpressed[5,10,13,28].

## Results

### PDAC distant metastases avidly consume glucose.
To address this possibility experimentally, we took advantage of a unique panel of clonal cell lines and tumor tissues collected from PDAC patients by rapid autopsies[5,11]. These samples have been heavily utilized by us[28,29] and others[5,10,12,13,18] to investigate traits that evolve in PDAC patients, since matched tumor tissues are available from the same individual patient(s) and the cell lines represent sequence-verified subclones that retain the morphologic, genetic, epigenomic, transcriptomic, and phenotypic properties of the parental tissues from which they were derived[5,10,11,13,28]. This included matched PGD-dependent liver and lung metastatic subclones that diverged from a PGD-independent metastatic peritoneal deposit in one patient (patient 38), a PGD-dependent primary tumor subclone that seeded distant metastasis in another patient (patient 13), matched liver and lung metastases from yet another patient (patient 2), and individual PGD-dependent metastases collected from additional patients[5,10,18,28,29].

In the rapid autopsy cohort, an intrinsic property of PGD dependence is constitutively elevated PGD catalytic rates (PGD$^{high}$)[29]. This results in steady-state depletion of the PGD substrate (6-phosphogluconate: 6PG)[28], indicating that provision of 6PG is rate limiting for high catalysis[29]. Consistent with this, 6PG was also the most depleted metabolite in a separate cohort of PGD-dependent cell lines reported in the cancer cell line

encyclopedia[30] (Supplementary Fig. 1a). Because 6PG is synthesized from glucose[29] and clinical experience with positron emission tomography imaging indicates that distant metastases avidly consume glucose in vivo[25], we hypothesized that PGD-dependent PDACs might have evolved intrinsic mechanism(s) that allowed them to consume the excess glucose required to support high PGD catalysis. To begin testing this hypothesis, we first verified that glucose consumption rates were recurrently elevated in the PGD-dependent subclones from the rapid autopsy cohort, as compared to a control panel of PGD-independent PDACs isolated from primary tumors (Supplementary Fig. 1b) and metastatic peritoneal deposits[28,29] (Fig. 1b). We next surveyed our previous RNA-sequencing (RNA-seq) datasets generated on a subset of these cells[28] to determine if any genes involved in glucose homeostasis might be dysregulated. From these data, we identified the TXNIP gene as recurrently suppressed in distant metastases. This finding was intriguing because TXNIP encodes a multifunctional protein that normally maintains glucose homeostasis by participating in glucose-sensing negative feedback loops that restrict excessive uptake (Fig. 1c)[31,32].

### TXNIP is recurrently suppressed in distant metastases.
To more rigorously evaluate TXNIP status in primary and metastatic pancreatic cancers, a comprehensive analysis of TXNIP expression was conducted across multiple sources of PDAC patient samples. We first confirmed that TXNIP transcripts were recurrently suppressed in the PGD-dependent rapid autopsy lines by quantitative reverse transcription PCR (RT-qPCR), as compared to PGD-independent controls (Fig. 1d). TXNIP was also suppressed in previously published serial analysis of gene expression (SAGE) datasets[13] that compared a subset of the PGD-dependent rapid autopsy metastases[5] to a separate cohort of 23 bulk primary tumors[13] (Supplementary Fig. 2a).

For a more in-depth analysis of in vivo TXNIP expression, we next evaluated 127 RNA-seq datasets that were recently generated on macrodissected tissue sections sampled from multiple regions of primary tumor ($n = 95$) that were matched with individual liver metastases ($n = 32$) within the same individual patient(s) of a separate rapid autopsy cohort ($n = 10$ total patients)[10,33]. Multiregional sequencing was employed for primary tumors because unlike the more uniform metastases, primary PDACs are a heterogeneous mixture of geographically distinct subclonal populations with divergent phenotypic traits[5,10,28,33]. As might be expected, TXNIP expression was heterogeneous among individual samples within the same patient(s) (Supplementary Fig. 2b), and baseline expression of pooled samples was also significantly variable between patients (Supplementary Fig. 2c; $p < 0.0001$ by one-way analysis of variance (ANOVA) and Kruskal-Wallis tests). However, in all ten patients the highest TXNIP expression was detected from a distinct region within the primary tumor ($p < 0.0001$ by two-sided Fisher's exact test), and despite regional heterogeneity the pooled primary tumor samples expressed significantly higher TXNIP overall than pooled liver metastases irrespective of whether samples were analyzed by baseline expression signals (Fig. 1e), after correction for the differences in baseline expression between patients (Fig. 1f), or after further corrections for tumor purity in a smaller subset of samples with available estimates[33] (Supplementary Fig. 2d). Suppression of TXNIP was also more directly observed as liver metastases diverged from the primary tumor using previously constructed phylogenetic trees[33] for a subset of patients with adequate sampling and tumor purities (Fig. 1g, Supplementary Fig. 2e–f), with immunohistochemical (IHC) confirmation on a subset of samples with available matched formalin-fixed tissue sections (Fig. 1g, right panels). Thus, heterogeneous TXNIP expression in

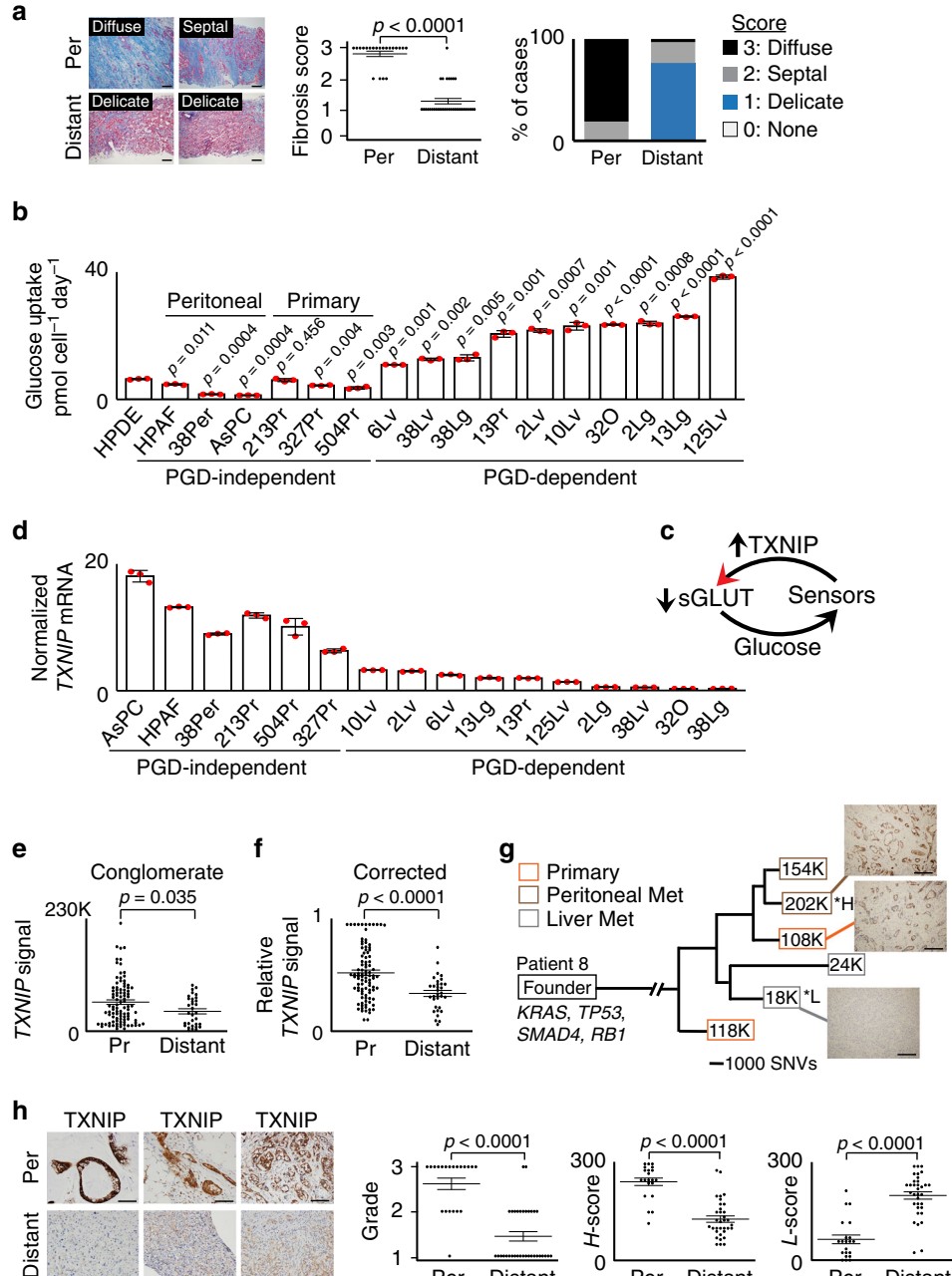

**Fig. 1 TXNIP is recurrently suppressed in PDAC distant metastases. a** (Left) Representative Masson's trichrome stains for collagen (blue) shows metastatic peritoneal deposits (top, $n = 21$ individual patient samples) embedded within diffuse geographic (scored 3) or thick septal (scored 2) fibrosis. Delicate thin strands of fibrosis (scored 1) are instead present in distant metastases (bottom, $n = 34$ individual patient samples). Scale bars: 200 μm. (Right) Graphical summaries of fibrosis scores (error bars: s.e.m., $p < 0.0001$ by two-tailed $t$ tests and two-sided Mann–Whitney $U$ tests). **b** Glucose consumption rates were elevated for the indicated PGD-dependent cells, relative to the indicated PGD-independent control cells ($n = 3$ technical replicates, error bars: s.d.m., indicated $p$ values calculated by two-tailed $t$ tests). **c** Illustration depicting how glucose-sensing negative feedback loops utilize TXNIP to prevent excessive glucose uptake. **d** *TXNIP* mRNA expression was downregulated in PGD-dependent subclones relative to PGD-independent controls ($n = 3$ technical replicates, error bars: s.d.m.). **e** *TXNIP* transcript levels were lower overall in liver metastases (distant, average: 40,790, $n = 32$ separate metastases) than from multiregional areas in primary tumors (Pr, average: 59,653, $n = 95$ separate primary regions). Data represent total *TXNIP* read counts from RNA-seq datasets ($n = 10$ total patients, error bars: s.e.m., $p = 0.035$ by two-sided Mann–Whitney $U$ tests). **f** Similar results were obtained when *TXNIP* read counts were corrected for differences in baseline expression ($n = 95$ primary tumor regions [Pr] and 32 liver metastases [Distant] from 10 total patients, error bars: s.e.m., $p < 0.0001$ two-sided Mann–Whitney $U$ tests). **g** Phylogenetic tree of patient 8 showing *TXNIP* expression for the indicated primary tumor subclones (orange boxes), metastatic peritoneal deposits (brown boxes), and liver metastases (gray boxes). Boxed values indicate *TXNIP* transcript expression from RNA-seq data (K units: thousands of read counts divided by the estimated tumor purity fractions, *H: highest, *L: lowest, SNV scale: single-nucleotide variants). IHC stains (connected by lines) confirmed loss of TXNIP protein in the indicated liver metastasis (scale bars: 200 μm). Similar results were obtained for two other patients with available phylogenetic data (Supplementary Fig. 1e, f). **h** (Left) Representative IHC stains for TXNIP (brown) show diffusely strong reactivity in peritoneal metastatic cells ($n = 21$ individual patient samples). Distant metastases show low or heterogeneous staining ($n = 34$ individual patient samples). Scale bars: 200 μm. Plots: IHC scored by a three-tiered grading scheme (left), *H*-scores weighted for high expression (middle), and *H*-scores weighted for low expression (right), as described in the "Methods" (error bars: s.e.m., $p < 0.0001$ by two-sided Mann–Whitney $U$ tests).

primary tumors largely resolved into a low expression state during hepatic distant metastasis.

Like the PDAC cell lines isolated from metastatic peritoneal deposits (Fig. 1d), *TXNIP* was also differentially expressed between a pair of matched liver and peritoneal metastases included in the RNA-seq analysis (Fig. 1g). These observations tentatively suggested that *TXNIP* status might further reflect divergent modes of metastatic spread. To more thoroughly test this possibility in vivo, we evaluated TXNIP protein expression by immunohistochemical (IHC) scoring of fixed tissue biopsies sampled from a separate collection of 34 unpaired distant metastases (32 liver, 2 lung; 30/34 untreated) and 21 unpaired metastatic peritoneal deposits (21/21 untreated). Compared with the nearly uniform high *TXNIP* expression in the peritoneal deposits, TXNIP was suppressed in most distant metastases, which we replicated using three separate histopathologic scoring systems (Fig. 1h, Supplementary Figs. 3 and 4) with high concordance (Pearson's $r = 0.938$, Supplementary Fig. 5). Based on the collective data acquired from a total of 235 cell and tissue samples spanning 112 patients from multiple independent cohorts evaluated by a variety of experimental assays, we conclude that TXNIP was recurrently suppressed in PDAC distant metastases.

**PGD suppresses TXNIP.** The association between PGD dependence, avid glucose consumption, and low *TXNIP* expression raised the intriguing possibility that high PGD catalysis might stimulate glucose uptake by suppressing TXNIP (Fig. 2a), especially since TXNIP restricts glucose import by promoting endocytosis of the glucose transporters GLUT1 (*SLC27A1*)[31] and GLUT4 (*SLC27A4*)[32] off the cell surface (Fig. 1c). Because *SLC27A1* (but not *SCL27A4*) was expressed in the rapid autopsy cells[28] and blockade of the encoded GLUT1 protein[34] recurrently slowed glucose consumption rates (Supplementary Fig. 6a), we performed a series of experiments to examine how PGD activation status might impact *TXNIP* expression, GLUT1 surface retention, and glucose consumption. Confocal immunofluorescence (IF) experiments first verified that matched PGD[high] liver and lung metastases expressed low cytosolic TXNIP with strong surface retention of GLUT1, as compared to a PGD[low] peritoneal metastasis from the same patient (Fig. 2b). To determine if those findings were PGD-dependent, we examined a PGD[high] subclone that was converted to PGD[low] with heterozygous Crispr/Cas mutants conferring 50% reductions in PGD catalytic activity[29]. This resulted in the upregulation of *TXNIP* transcripts with concordant slowing of glucose consumption rates (Fig. 2c). Confocal IF experiments further demonstrated that these findings were coupled to loss of GLUT1 from the cell surface (Fig. 2d), and this effect was rescued by preventing TXNIP upregulation with RNA interference (RNAi) knockdown (Fig. 2e). As a final test of generality, pharmacologic inhibition of high PGD catalysis with 6-aminonicotinamide (6AN)[28,35] across the larger cohort of PGD-dependent subclones recurrently upregulated *TXNIP* transcripts (Fig. 2f), removed GLUT1 off of cell surfaces (Supplementary Fig. 6b), and slowed glucose consumption rates (Fig. 2f). Based on these collective data, we conclude that PGD stimulates glucose import by suppressing TXNIP in PGD-dependent PDACs (Fig. 2a).

**TXNIP opposes PGD.** The recurrent nature of the observations presented above (Figs. 1 and 2) raised the possibility that suppression of TXNIP might reflect an adaptation that was beneficial for metastasis. However, loss of TXNIP disrupts negative feedback controls that are important to maintain proper glucose homeostasis[31,32], raising questions as to what fitness advantages

might be gained. Because PDAC distant metastases can route excess glucose into metabolic pathways that fuel high PGD catalysis[29], we postulated that suppression of TXNIP might help support PGD activation by allowing glucose consumption rates to rise (Fig. 2a). If so, we hypothesized that restoring TXNIP into a PGD[high] background would phenocopy the effects of PGD inactivation. To this end, we expressed an exogenous TXNIP transgene within a PGD[high] distant metastasis to levels comparable with endogenous *TXNIP* expression in a matched PGD[low] peritoneal metastasis from the same patient (Fig. 3a). Like PGD inactivation (Fig. 2c–f), exogenous TXNIP successfully (re)-localized GLUT1 off the cell surface (Fig. 3a) with concordant reductions in glucose consumption (Fig. 3b). Although exogenous TXNIP did not influence PGD expression (Supplementary Fig. 7a, b), it significantly lowered PGD catalytic rates in PGD[high] cells (Fig. 3c). Similar reductions in PGD catalysis were observed during GLUT1 blockade (Supplementary Fig. 7c) and glucose starvation (Supplementary Fig. 7d), suggesting that the effects of TXNIP on PGD activity (Fig. 3c) were likely an indirect consequence of reduced glucose import (Fig. 3a, b). Exogenous TXNIP also phenocopied the growth defects observed during PGD inactivation[28,29], as it did not impair 2D proliferative growth (Supplementary Fig. 7e), yet strongly blocked 3D tumoroid growth of PGD[high] cells with no effect on the matched PGD[low] controls (Fig. 3d). Thus, restoring TXNIP suppressed PGD-dependent phenotypic properties. Because these findings implied that loss of TXNIP was an important step for acquisition of PGD dependence, we next explored in more detail potential mechanisms whereby PGD might suppress TXNIP in PDAC distant metastases.

**PGD prevents transcriptional activation of TXNIP.** Glucose homeostasis is normally maintained by nutrient-sensing negative feedback loops that utilize TXNIP to prevent excessive uptake when external glucose supplies are replete (Fig. 1c)[31,36]. How then is TXNIP suppressed in distant metastases when glucose is both replete and avidly consumed? Because PGD repressed TXNIP at both the protein and messenger RNA levels, we hypothesized that *TXNIP* transcription might be impaired in PGD-dependent subclones. MondoA (*MLXIP*) and ChREBP (*MLXIPL*) are cytosolic MLX-interacting transcription factors that sense glucose-derived metabolites through unknown mechanisms[36,37]. Under glucose-replete conditions, these factors normally traffic from the cytosol to the nucleus where they transcriptionally activate glucose-responsive genes, including *TXNIP*[37]. Of these two, we focused on MondoA since it was expressed in our RNA-seq datasets[28]. Confocal IF experiments first verified that MondoA was concentrated in the nucleus of PGD[low] control cells, as expected under glucose-replete conditions (Fig. 4a, left panel). In contrast, MondoA was more diffusely distributed between the nuclear and cytosolic compartments in PGD[high] subclones from the same patient (Fig. 4a, right panels), with corresponding reductions in MondoA binding to the *TXNIP* promoter by chromatin immunoprecipitation (ChIP) (Fig. 4b). Inactivation of PGD[high] with Crispr/Cas also concentrated MondoA back into the nucleus (Fig. 4c) with ChIP enrichments at the *TXNIP* promoter (Fig. 4d), indicating that PGD[high] might regulate TXNIP and GLUT1 through MondoA. That impression was confirmed by MondoA knockdown during PGD inactivation, which strongly rescued both TXNIP suppression and surface GLUT1 retention (Fig. 4e).

PGD[high] catalysis (over)consumes intracellular 6PG to low steady-state concentrations[28]. Conversely, PGD inhibition causes 6PG to accumulate[28]. Because of this, we further hypothesized that PGD activity might indirectly influence MondoA trafficking

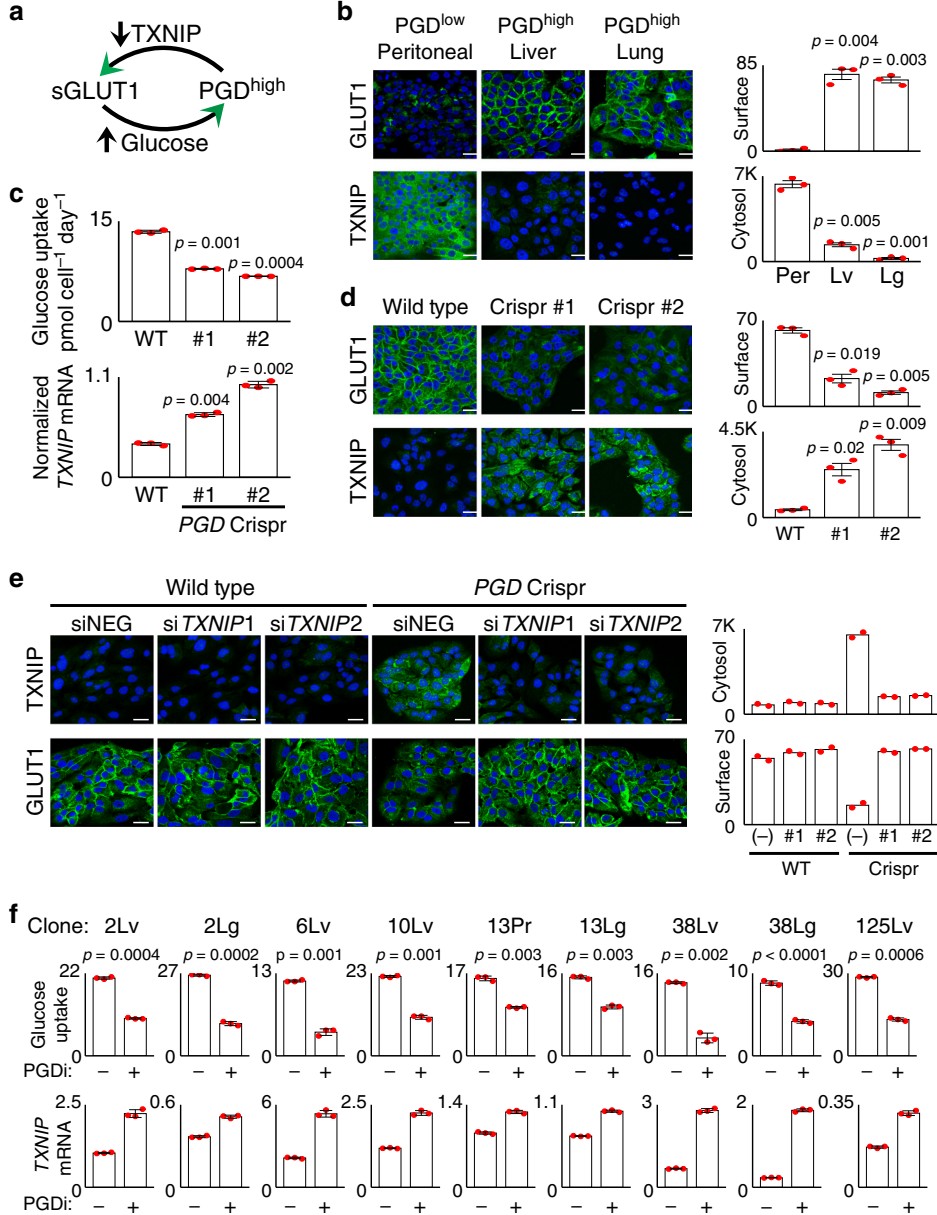

**Fig. 2 TXNIP suppression is PGD-dependent. a** Illustration depicting how PGD-driven suppression of TXNIP can allow retention of surface GLUT1 with increased glucose uptake. **b** Confocal IF detected high surface GLUT1 signals (top, cobblestone pattern) and reduced cytosolic TXNIP signals (bottom) in PGD[high] liver and lung metastases compared with a control PGD[low] peritoneal deposit from the same patient. **c** PGD Crispr/Cas[5] (#1, #2) reduced glucose consumption rates (top) and increased *TXNIP* expression (bottom qRT-PCR) relative to wild-type (WT) baseline controls ($n = 3$ technical replicates per sample, error bars: s.d.m., indicated $p$ values calculated by two-tailed $t$ tests). **d** Confocal IF detected reductions in surface GLUT1 (top) with increased cytosolic TXNIP (bottom) in response to PGD Crispr/Cas (#1, #2), compared to wild-type baseline controls (WT). **e** Similar confocal IF experiments show that siRNAs against *TXNIP* (#1, #2) prevents upregulation of TXNIP protein and rescues surface GLUT1 localization during PGD inactivation (*PGD* Crispr/ Cas, right three panels), as compared to both nontargeting control siRNAs (siNEG) and wild-type control cells (left three panels) ($n = 2$ biological replicates, error bars: s.e.m.). **f** Treatment of the indicated PGD-dependent subclones with PGD inhibitor (PGDi: 50 µM 6AN) increased *TXNIP* expression by RT-qPCR (bottom) and slowed of glucose consumption rates (top) ($n = 3$ technical replicates per sample, error bars: s.d.m., indicated $p$ values for glucose consumption rates calculated by two-tailed $t$ tests). For all IF experiments (unless indicated otherwise): green, antibody signal; blue, Hoechst signal; $n = 3$ biological replicates, error bars: s.e.m., indicated $p$ values calculated by two-tailed $t$ tests, scale bars: 20 µm.

through changes in intracellular 6PG concentration(s), similar to how other glucose-derived metabolites are proposed to influence MondoA[36]. To begin testing this, we first verified that inhibiting PGD in PGD-dependent PDACs caused 6PG (but not G6P[29]) to accumulate with corresponding increases in MondoA nuclear–cytosolic ratios (Supplementary Fig. 8a, b). MondoA was concordantly enriched at the *TXNIP* promoter by ChIP (Fig. 4f), and the resulting increases in *TXNIP* transcripts closely matched the

kinetics of 6PG accumulation (Supplementary Fig. 8c). To more directly determine if 6PG could influence MondoA trafficking, we raised intracellular 6PG to concentrations beyond what PGD could acutely consume at baseline by treating PGD[high] cells with excess (20 mM) exogenous 6PG (Fig. 4g, top right panel). Remarkably, 6PG (but not G6P) strongly concentrated MondoA back into the nucleus, and this effect was fully rescued by depleting the excess intracellular 6PG with PGD overexpression

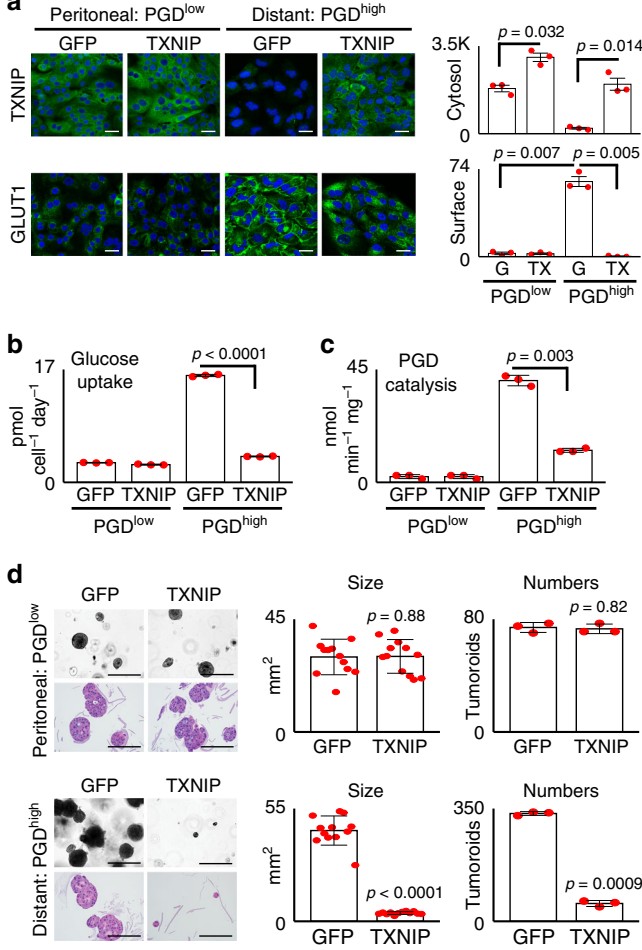

**Fig. 3 Exogenous TXNIP phenocopies PGD inactivation. a** Expression of exogenous TXNIP (top panels) in the PGD^high distant metastasis (38Lg) to levels comparable with baseline *TXNIP* expression in control PGD^low peritoneal deposit (38Per) removed GLUT1 from the cell surface in PGD^high cells (bottom panels) (*n* = 3 biological replicates; error bars: s.e.m., indicated *p* values calculated by two-tailed *t* tests; G: GFP; TX: TXNIP). Scale bars: 20 μm. **b** Exogenous TXNIP slowed glucose consumption in the PGD^high cells down to rates comparable with matched PGD^low controls (*n* = 3 technical replicates, error bars: s.d.m., *p* < 0.0001 by two-tailed *t* tests). **c** Exogenous TXNIP also slowed PGD catalytic rates in PGD^high cells (*y*-axis: NADPH production rates; *n* = 3 technical replicates, error bars: s.d.m., *p* = 0.003 by two-tailed *t* tests). **d** Exogenous TXNIP strongly impaired 3D tumoroid growth of PGD^high cells with no effect on control PGD^low controls (*n* = 3–12 technical replicates; error bars: s.d.m., indicated *p* values calculated by two-tailed *t* tests). Top panels: brightfield microscopy of live tumoroids. Bottom panels: H&E stains of fixed tumoroids. Scale bars: 400 μm.

(Fig. 4g). Based on these collective data, we conclude that PGD^high can redirect MondoA trafficking to prevent transcriptional upregulation of *TXNIP*. Although defining the precise physical mechanisms of how MondoA senses 6PG are well beyond the scope of the current work, we speculate that MondoA monitors 6PG as a surrogate of intracellular glucose in this setting. If so, then depletion of 6PG[28] by abnormally high PGD catalysis[29] would mimic glucose starvation and redirect MondoA traffic away from *TXNIP* even as glucose consumption rates rise to excessive levels.

**PGD and glucose cooperate to reprogram chromatin.** Our previous metabolomics studies indicated that PGD-dependent

PDACs can route glucose into multiple pathways[28,29], raising the possibility that excess glucose could be used to fuel other select-able traits beyond the PGD reaction itself. Histones within gene regulatory elements and large chromatin domains are globally reprogrammed into a hyperacetylated state during PDAC distant metastasis, and these events are permissive for activation of the metastatic transcriptome[28,38]. Because glucose is replete along metastatic routes and global histone acetylation is generally more sensitive to glucose than other acetyl-donor nutrients[39], we hypothesized that PGD-driven glucose uptake might facilitate hyperacetylation of metastatic chromatin.

Based on this hypothesis, we predicted that histone hyper-acetylation was maintained by a glucose-fueled pathway consist-ing of PGD (for *TXNIP* suppression), surface GLUT1 (for glucose import), and ACLY (for conversion of glucose-derived citrate into acetyl CoA[39,40]) (Fig. 5a). Inhibiting any step of this pathway should therefore result in quantitative reductions in histone acetylation that are detectable at the global level within nuclei. Consistent with those predictions, glucose deprivation, PGD inhibition, GLUT1 inhibition[34], and ACLY inhibition[39] all reversed global H3K27Ac in a PGD^high distant metastasis down to levels comparable with a matched PGD^low peritoneal metastasis from the same patient by confocal IF imaging of nuclei (Fig. 5b). Those results were supported by RNAi knock-down of ACLY, which also lowered global H3K27Ac and H4K16Ac (Fig. 5c). As a more comprehensive test of generality, identical glucose deprivation and inhibitor experiments were conducted across the full set of PGD-dependent subclones from the rapid autopsy cohort. Each intervention consistently lowered H3K27Ac across all test samples (Fig. 5d), indicating that each component of the pathway was required to maintain global acetylation both within and between patients.

Our hypothesis further implied that suppression of TXNIP should be required to maintain histone hyperacetylation (Fig. 5a). Consistent with this, RNAi knockdown of either endogenous TXNIP or *MLXIP*/MondoA (which transcriptionally activates *TXNIP*) rescued loss of H3K27Ac and H4K16Ac during PGD inactivation (Fig. 6a). Likewise, restoring TXNIP exogenously reversed histone hyperacetylation in PGD^high cells down to levels comparable with PGD^low controls from the same patient, as visualized in bulk nuclei by confocal IF (Fig. 6b) and in acid-extracted bulk histones by immunoblots (Supplementary Fig. 9a). Concordant with the quantitative reductions in global acetylation[28,38], exogenous TXNIP also quantitatively repressed 11/15 PGD-dependent genes[28] from a panel designed to reflect malignant properties enriched in PDAC distant metastases[17,28,38] without nonspecific reactivation of silenced genes (Fig. 6c, Supplementary Fig. 9b). Native ChIP assays targeting euchromatic gene regulatory elements (Fig. 6d, e) and large chromatin domains (Fig. 6f) verified that the global reductions in H3K27Ac were targeted to hyper-acetylated regions encoding prometastatic genes that were down-regulated by TXNIP (*ODC1*, *FOXA1*[38], and *CDH2*[28]). The results presented in Figs. 5 and 6 collectively suggest that PGD-driven suppression of TXNIP helps facilitate glucose-fueled hyperacetylation of malignant PDAC chromatin (Fig. 5a).

**PGD and TXNIP regulate experimental metastasis.** The low passage PGD^high clonal cell lines examined in this study all possess brisk 3D tumorigenic capacity in vitro[28,29] and retain phenotypic characteristics of the tissues from which they were derived[5,10,11,13,28]. However, in vivo experimental metastasis assays have thus far been hampered because these cells do not efficiently metastasize when injected into the bloodstream of immunodeficient mice[28]. To circumvent this limitation while remaining within the scope of the well-characterized rapid autopsy samples under study, we tested an additional rapid

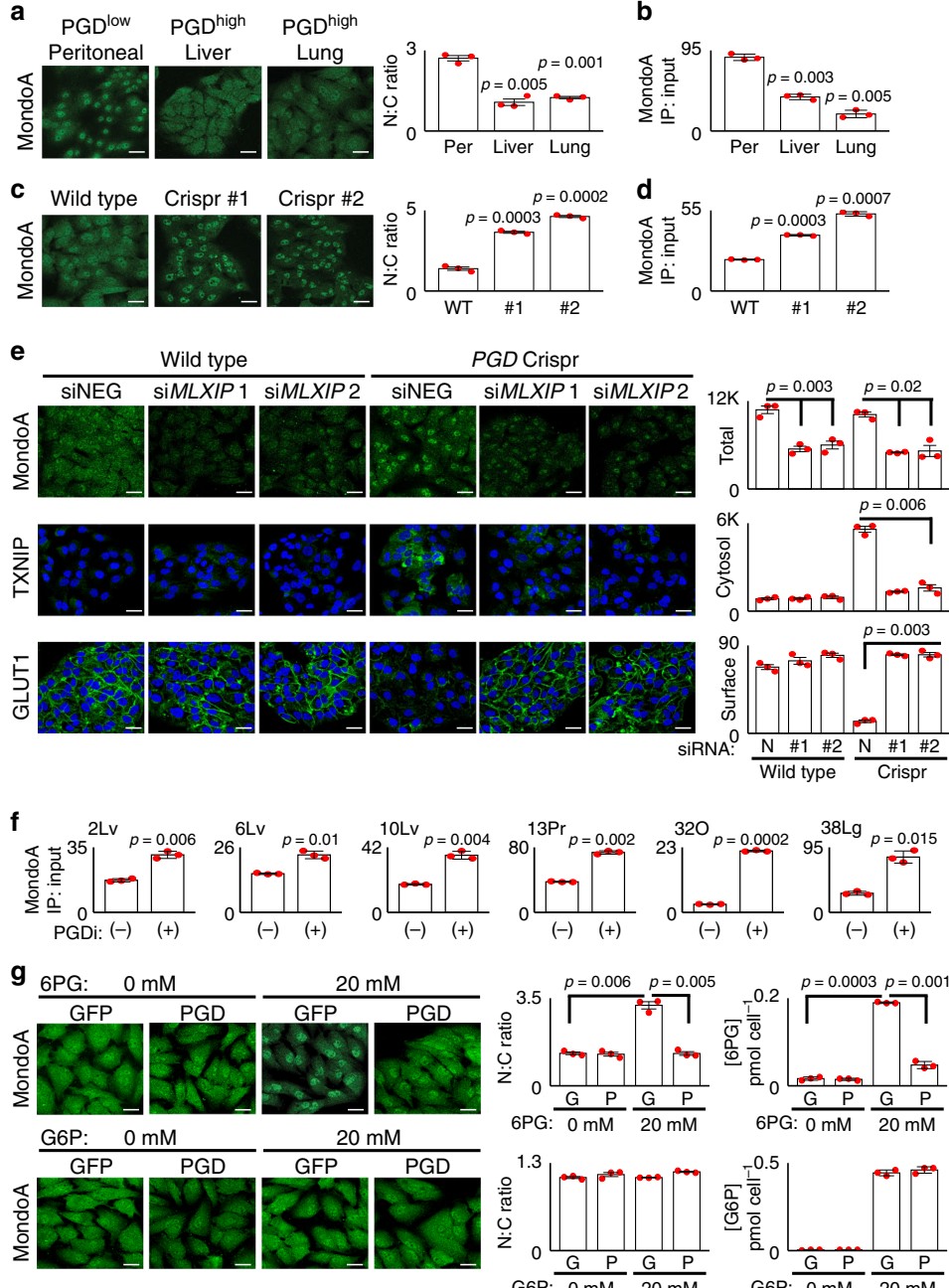

**Fig. 4 PGD status influences MondoA trafficking. a** MondoA was concentrated in the nucleus of PGD^low 38Per peritoneal control cells by IF (left panel). In contrast, MondoA was distributed more equally between the nuclei and cytosol in PGD^high liver and lung metastases from the same patient (38Lv, 38Lg). **b** ChIP assays also detected reduced MondoA enrichments at the *TXNIP* promoter in the liver and lung metastases. **c** PGD inactivation by Crispr/Cas (sgRNAs #1, #2) concentrated MondoA into the nucleus of PGD^high 38Lg cells, as compared to wild-type controls. **d** PGD Crispr/Cas also increased MondoA enrichments at the *TXNIP* promoter by ChIP assays in 38Lg. **e** siRNAs targeting MondoA (si*MLXIP* #1, #2) knocked down MondoA in 38Lg cells (top panels) and prevented upregulation of TXNIP during Crispr/Cas PGD inactivation (middle panels). MondoA knockdown also rescued surface GLUT1 (bottom panels). **f** ChIP assays detected increased MondoA enrichments at the *TXNIP* promoter across the indicated PGD^high subclones treated with PGD inhibitor (PGDi: 6AN). **g** (Top panels) Treatment of GFP-expressing PGD^high 38Lg cells with excess (20 mM) 6PG concentrated MondoA in the nucleus by IF. This effect was rescued by depleting the excess 6PG with PGD overexpression[4]. Measurements of intracellular 6PG concentrations (right plots) confirmed that 6PG accumulated in treated GFP-expressing cells and that the excess was depleted in exogenous PGD-(over)expressing cells. Bottom panels: G6P control treatments failed to concentrate MondoA into the nucleus under any condition. For IF experiments: green, antibody signal; blue, Hoechst signal; $n = 3$ biological replicates, error bars: s.e.m., indicated $p$ values calculated by two-tailed $t$ tests, scale bars: 20 μm. For ChIP experiments: $n = 3$ technical replicates, error bars: s.d.m., indicated $p$ values calculated by two-tailed $t$ tests, results are representative of two biological replicates.

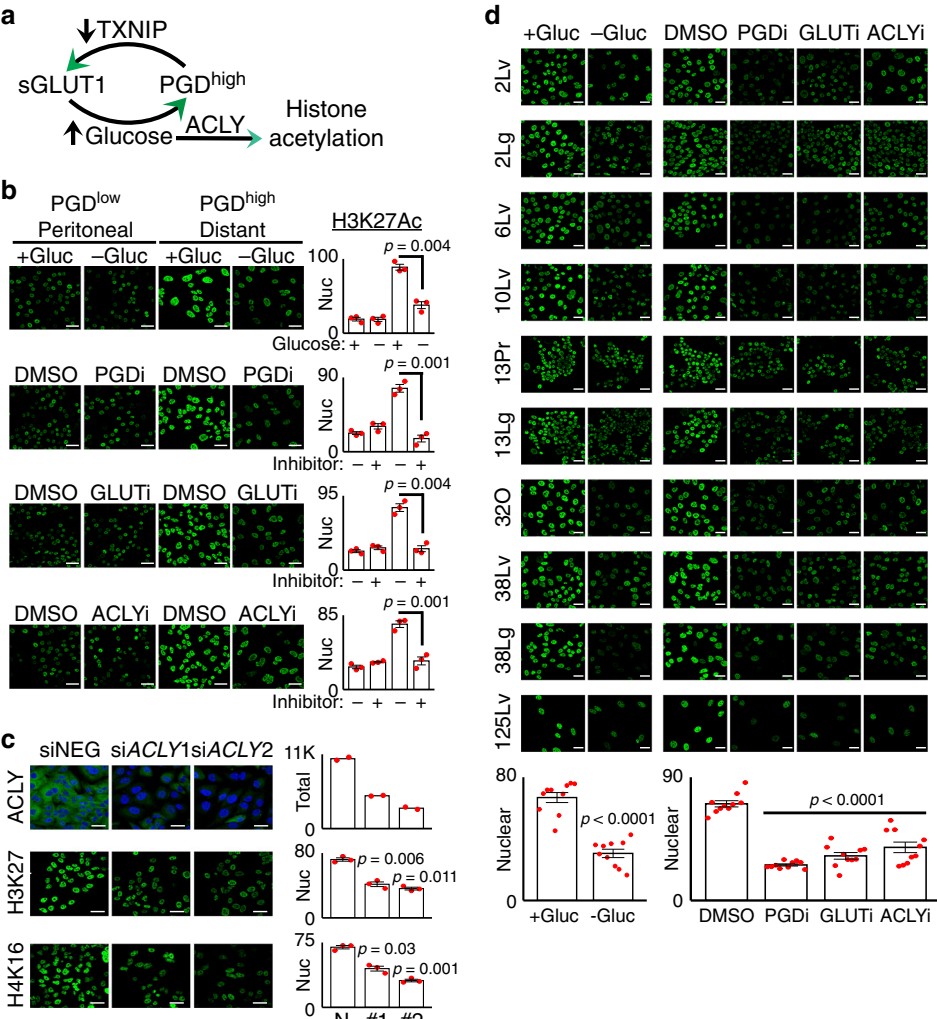

**Fig. 5 A PGD-GLUT1-ACLY pathway hyperacetylates metastatic chromatin. a** Illustration of how PGD-driven retention of surface GLUT1 can provide excess glucose for ACLY-dependent histone hyperacetylation. **b** Glucose deprivation (−Gluc, top row), PGD inhibition (PGDi, second row), GLUT1 inhibition (GLUTi, third row), and ACLY inhibition (ACLYi, bottom row) all reduced global H3K27Ac in the nuclei of PGD^high 38Lg cells (right panels) down to levels comparable with PGD^low 38Per controls from the same patient (left panels) by confocal IF. **c** Treatment of PGD^high 38Lg cells with siRNAs against ACLY (#1, #2) knocked down ACLY expression (top panels, $n = 2$ biological replicates) and reduced histone acetylation (H3K27Ac: middle panels;, H4K16Ac: bottom panels; $n = 3$ biological replicates) relative to nontargeting siRNA controls (siNEG). **d** Glucose deprivation, PGDi, GLUT1i, and ACLYi recurrently lowered H3K27Ac across the full complement of PGD^high subclones (left-hand labels). Quantified nuclear H3K27Ac values for all the indicated experiments pooled together are plotted at the bottom. For all IF experiments (unless otherwise indicated): green, antibody signal, $n = 3$ biological replicates, error bars: s.e.m., indicated $p$ values calculated by two-tailed $t$ tests, scale bars: 20 μm.

autopsy PGD^high line that was instead derived from a primary tumor subclone that was genetically[5] and phenotypically[28,29] similar to distant metastases in that patient (13Pr). Because this line was isolated from the primary tumor rather than an established metastatic tumor, we hypothesized that it might have retained a fuller spectrum of the metastatic cascade. Consistent with this, athymic nude mice developed grossly visible liver metastases following intrasplenic implantation of 13Pr cells (Fig. 7a, left panels gross, $n = 4/4$ mice; range 2–15 metastatic tumors per mouse). Microscopically, the lesions were centered within the interior of the hepatic parenchyma (Fig. 7a, left panels, hematoxylin and eosin (H&E)), which is consistent with hematogenous seeding as seen in patients with distant metastasis. The pancreatic parenchyma was uninvolved ($n = 0/4$). In contrast, intrasplenic implantation of control PGD^low cells (isolated from a metastatic peritoneal deposit) resulted in the development of grossly evident pancreatic tumors ($n = 5/5$ mice, Fig. 7a, right panels). Although the intrahepatic parenchyma was not involved,

careful examination identified small tumor deposits implanted on the outside surface of the liver (Fig. 7a, right panels, $n = 3/5$ mice), which is consistent with non-hematogenous seeding as seen in patients with peritoneal carcinomatosis.

We therefore utilized 13Pr to formally test if intrahepatic metastasis was dependent on PGD and/or suppression of TXNIP in vivo. To this end, we targeted PGD for inactivation with Crispr/Cas (Supplementary Fig. 10a) and restored TXNIP downstream of PGD through exogenous transgene expression. Importantly, both interventions reproduced the phenotypic spectrum of PGD inactivation[28,29] (Figs. 2–6), including TXNIP upregulation with removal of surface GLUT1 (Fig. 7b, c), slowing of glucose consumption rates (Supplementary Fig. 10b, c), loss of histone hyperacetylation (Fig. 7b, c), and impaired 3D tumoroid growth in vitro (Fig. 7d, e). PGD inactivation also increased MondoA nuclear–cytosolic ratios with corresponding MondoA enrichments at the *TXNIP* promoter (Supplementary Fig. 10d, e). Having verified that 13Pr was a suitable PGD-dependent host, we

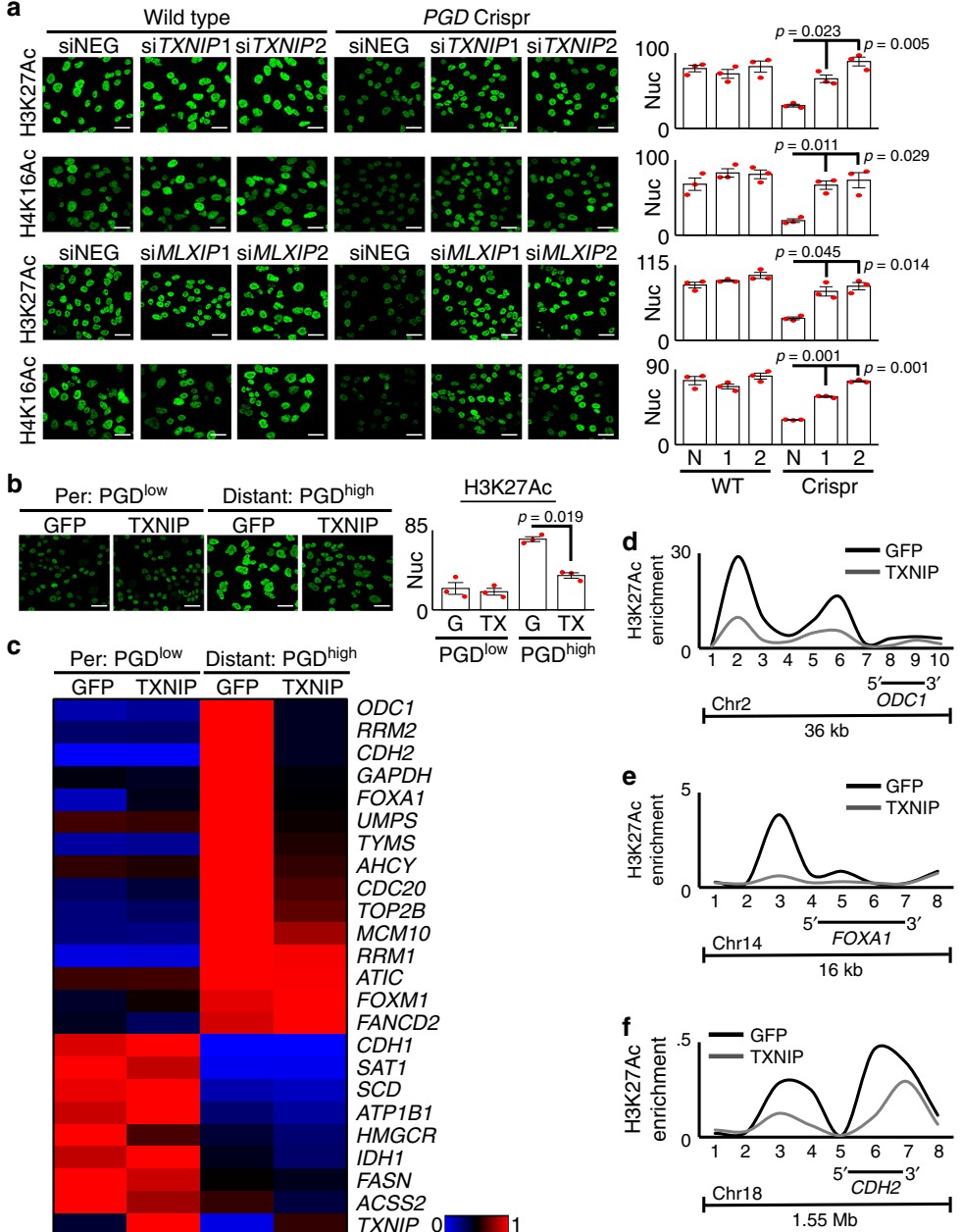

**Fig. 6 PGD-driven suppression of TXNIP maintains hyperacetylated chromatin. a** RNAi knockdown of either TXNIP (top two rows: si*TXNIP*1, 2) or MondoA (bottom two rows: si*MLXIP*1, 2) rescued loss of H3K27Ac and H4K16Ac in PGD$^{high}$ 38Lg cells during PGD inactivation (right three columns: *PGD* Crispr). **b** Expression of exogenous TXNIP in PGD$^{high}$ 38Lg cells (right two panels) reduced H3K27Ac to levels comparable with PGD$^{low}$ 38Per controls (left two panels). G, GFP; TX, TXNIP. **c** A heat map summarizing RT-qPCR data shows that recurrently upregulated genes in PDAC distant metastases (top 15, red/hot under Distant: PGD$^{high}$ GFP) were downregulated in response to exogenous TXNIP in PGD$^{high}$ 38Lg cells. A control panel of other downregulated genes (bottom, blue/cold under Distant: PGD$^{high}$ GFP) were not reactivated ($n = 3$ technical replicates, $p$ values calculated by two-sided $t$ tests are shown in Supplementary Fig. 9b and in the Source Data file). **d–f** Exogenous TXNIP also lowered high H3K27Ac enrichments across loci encoding *ODC1* (**d**), *FOXA1* (**e**), and *CDH2* (**f**) genes in PGD$^{high}$ cells by native ChIP assays. The $x$-axis indicates primer locations relative to the chromosome locations depicted underneath each plot (data is representative of 2 biological replicates). For all IF experiments: green, antibody signal, $n = 3$ biological replicates, error bars: s.e.m., indicated $p$ values calculated by two-tailed $t$ tests, scale bars: 20 µm.

conducted a new series of intrasplenic implantation experiments using 13Pr-derived engineered lines. As expected, athymic mice implanted with control 13Pr cells developed liver metastases ($n = 2/3$, Fig. 7f, left panels). In contrast, liver metastases failed to develop in athymic mice implanted with 13Pr cells expressing exogenous TXNIP ($n = 0/6$, Fig. 7f, middle panels), which matched the effects of PGD loss of function ($n = 0/6$, Fig. 7f, right panels). Thus, restoring TXNIP into 13Pr phenocopied the

effects of PGD inactivation, including prevention of experimental metastasis in vivo.

## Discussion

Here we report a malignant metabolic adaptation that allows human pancreatic cancers to consume abnormally high amounts of glucose, achieve PGD$^{high}$ status, and reprogram chromatin for metastasis. This adaptation was recurrently detected across the

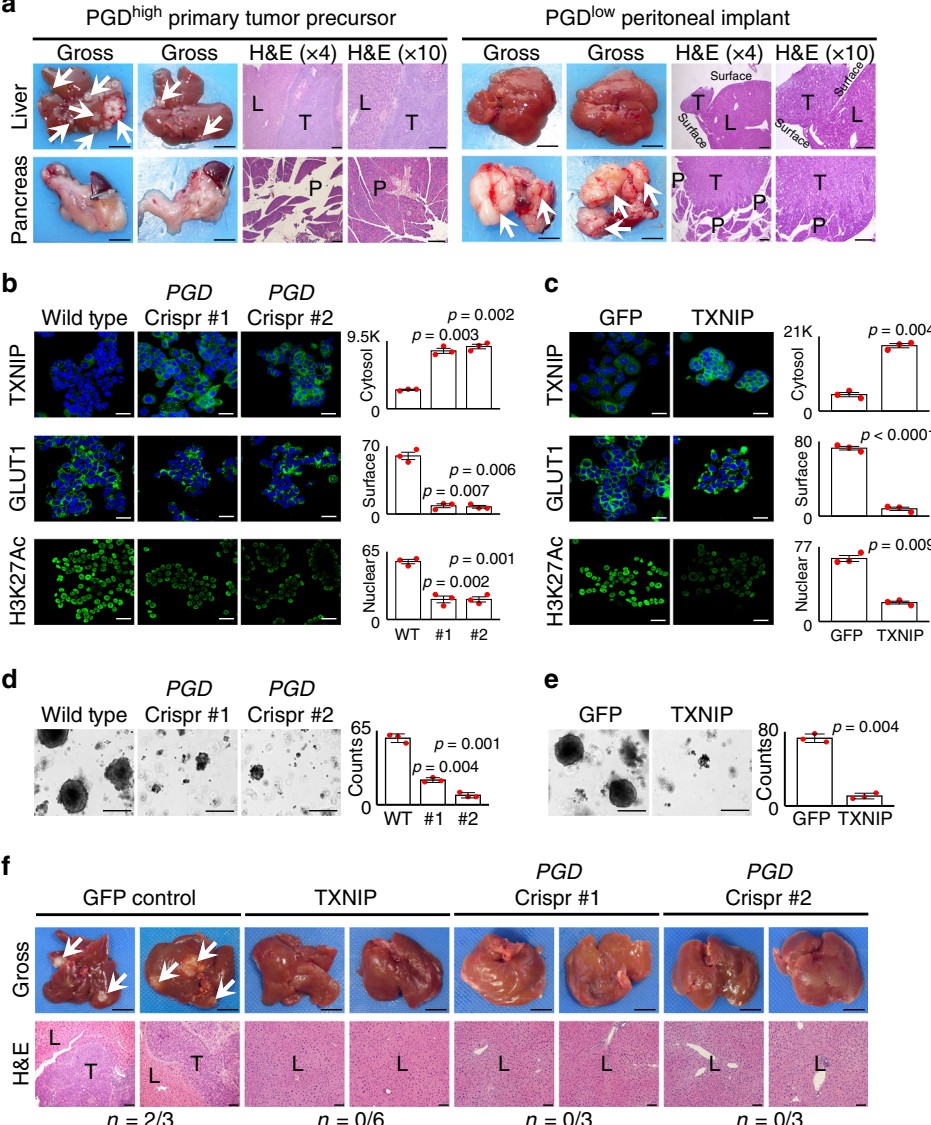

**Fig. 7 Experimental metastasis is dependent on PGD and suppression of TXNIP in vivo. a** (Left) PGD[high] 13Pr cells implanted into the spleens of athymic mice preferentially develop liver metastasis (arrows, $n = 4/4$) without involvement of the pancreatic parenchyma ($n = 0/4$), as shown by representative gross photographs and H&E-stained sections. (Right) In contrast, PGD[low] 38Per cells formed grossly evident pancreatic tumors ($n = 5/5$), with small tumor implants on the outside surface of the liver as seen on H&E-stained sections ($n = 3/5$). Gross photograph scale bars: 1 cm, H&E scale bars: 400 μm. L liver, P pancreas, T tumor. **b, c** Engineering 13Pr cells with PGD Crispr/Cas sgRNAs (**b**) or exogenous TXNIP (**c**) reversed TXNIP suppression (top), surface GLUT1 retention (middle), and H3K27 hyperacetylation (bottom) by confocal IF ($n = 3$ biological replicates, error bars: s.e.m., indicated $p$ values calculated by two-tailed $t$ tests, scale bars: 20 μm). **d, e** 13Pr cells engineered with Crispr/Cas PGD inactivating sgRNAs (**d**) or exogenous TXNIP (**e**) impaired 3D in vitro tumoroid growth relative to 13Pr control cells ($n = 3$ technical replicates, error bars: s.d.m., indicated $p$ values calculated by two-tailed $t$ tests, scale bars: 400 μm). **f** Representative gross and H&E-stained sections demonstrate that 13Pr cells engineered with exogenous TXNIP ($n = 0/6$) or Crispr/Cas inactivation of PGD ($n = 0/6$) failed to develop liver metastases compared to 13Pr control cells ($n = 2/3$). Gross photograph scale bars: 1 cm, H&E scale bars: 400 μm. L liver parenchyma, T tumor.

rapid autopsy patient samples examined here, in the apparent absence of a clear genetic underpinning[5,10]. Mechanistically, elevated glucose import supports high PGD catalysis, and PGD conversely supports elevated glucose uptake by interfering with MondoA-mediated transcriptional activation of *TXNIP*. While the products of PGD catalysis (NADPH, ribulose) are themselves protumorigenic, the excess glucose helps fuel additional prometastatic traits beyond the PGD reaction, including histone hyperacetylation with corresponding upregulation of malignant gene transcripts[28]. In the context of pancreatic cancer, this adaptation is more precisely defined as a metastable metaboloepigenetic program that requires a rich source of external glucose

to remain activated. Such reservoirs can be found all along the distant metastatic route, including unusually well-vascularized regions of the primary tumor, within the circulation, or at the metastatic site itself.

Several important questions remain that warrant further investigation. First, the PGD-driven adaptation described here could functionally engage with other prometastatic inputs. Possibilities include antioxidant defenses[28,41–44], energy-balancing pathways[31,45,46], other metabolites[47–50], nucleotide and reductive biosynthesis[28], acetylation-permissive conditions[51,52], transcription factor targeting[38,53], and nonhistone acetylation (including PGD itself[54]). Although the individual components of this adaptation are

not genetically altered, it remains a formal possibility that pre-existing genetic drivers[26,55,56], genome instability[18,20,57], and subtle or low-frequency genetic hits[4,5,17] could nonetheless influence the origins or evolutionary trajectories of this process. It will also be important to evaluate if our findings apply to other experimental settings outside the rapid autopsy patient cohort(s), including genetically engineered mouse models of PDAC. Although there appears to be selective pressure against TXNIP for distant metastasis, its roles during primary tumor growth and peritoneal metastasis remain undefined. Perhaps the most fundamental unanswered questions include the precise mechanism(s) of how PGD dependence initially emerges during subclonal evolution, and how each of the PGD-driven phenotypic properties are selected and refined thereafter[28,29,58,59].

More broadly, our findings raise the possibility of prometastatic positive feedback loops that eliminate negative feedback opposition (Figs. 1c and 2a). We speculate that this represents a meta-bolic feedback exchange strategy that cooperates with pre-existing genetic drivers to promote or even accelerate disease progression. Because exchange of TXNIP for PGD is a glucose-fueled, self-reinforcing process that regulates global epigenetic state (Fig. 5a), we envision that such strategies are devised to provide malignant cells with heritable metastatic adaptations[28,50,60–64] that parlay niche-refined nutrient reservoirs into selectable traits without genetic constraints. In this way, metabolically favorable clonal expansions can continue to occur late into disease evolution[5] through epigenetic mechanisms[28] after selection of genetic drivers is complete[4,5,10]. Our data indicate that clonal exchange of TXNIP for PGD supports distant metastasis in pancreatic cancer patients. If similar metaboloepigenetic programs are prevalent in advanced cancers, then interventions that interfere with these vulnerabilities could provide therapeutic benefits even for patients with widely metastatic disease.

## Methods

**Reagent sources**. The rapid autopsy cell lines are previously described[5,28,29]. AsPC-1, HPAF-II, and primary PDAC lines[65] were purchased from ATCC. Human pancreatic duct epithelial (HPDE) cells were kindly provided by Dr. Dan Beauchamp. All primer sequences and antibodies are listed in Table 1 of the Supplementary Information file.

**Staining of patient tissue samples**. Archived H&E-stained slides, trichrome-stained slides, and paraffin-embedded blocks with residual formalin-fixed tissue from metastases biopsied or resected at Vanderbilt between 2013 and 2018 were identified and collected. IHC stains for TXNIP were performed in the Vanderbilt Translational Pathology Core Research (TPSR) facility under standard conditions using a 1:250 antibody dilution. Appropriately stained quality control tissues were included with each experimental batch. Studies were approved by the Vanderbilt Institutional Review Board.

**Histopathology**. H&E, Masson's trichrome, and IHC stains were evaluated for tumor cells, stroma content, and tumor cell IHC staining by a board-certified anatomic pathologist that specializes in gastrointestinal, liver, and pancreatico-biliary pathology (O.G.M.). Fibrosis was scored three (diffuse) if tumor glands were widely separated by geographic swaths of fibrosis (similar to a scar), two (septal) if tumor glands were separated by fibrous bands (similar to cirrhotic bridging fibrosis), and one (delicate) if tumor glands were separated by thin fibrous strands (similar to steatohepatitic pericellular fibrosis). No cases met the criteria for zero (absent) fibrosis. IHC stains were graded according to a typical three-tiered scoring scheme. Stains were scored three (high) if ≥80% of tumor cells were strongly positive, two (moderate or heterogeneous) if ≥80% of tumor cells were moderately positive or if staining intensities were variable, and one (low) if ≥80% of tumor cells were weakly positive or negative. Semiquantitative values were obtained by applying the H-scoring system to IHC stains. H-scores weighted for high expression were calculated according to the formula: $0 \times$ (%cells with zero intensity) $+ 1 \times$ (%cells with 1+ intensity) $+ 2 \times$ (%cells with 2+ intensity) $+ 3 \times$ (%cells with 3+ intensity). H-scores weighted for low expression were calculated according to the formula $0 \times$ (%cells with 3+ intensity) $+ 1 \times$ (%cells with 2+ intensity) $+ 2 \times$ (%cells with 3+ intensity) $+ 3 \times$ (%cells with zero intensity). Four peritoneal control PDACs from the stroma analysis had insufficient tissue for IHC. These were replaced with peritoneal metastases from appendiceal mucinous primaries, since these tumors preferentially metastasize to the peritoneal cavity.

**Cell culture**. Sequence-verified low passage (2–15) rapid autopsy cells were cultured at 37 °C in DMEM (Dulbecco's modified Eagle's medium) supplemented with 10% fetal bovine serum (Gibco) and 1× glutamax (Gibco), free of mycoplasma (Sigma Lookout). Cells were uniformly cultured under high (10–25 mM) glucose conditions, unless otherwise stated. RNAi experiments were performed with small interfering RNA (siRNA) transfections (Oligofectamine, Life)[3,4] using negative control siRNA oligonucleotides (Sigma, SIC002) and siRNAs oligonucleotides targeting TXNIP (Sigma: SASI_Hs01_00184126 and SASI_Hs01_00184127) or MondoA/MLXIP (Sigma: SASI_Hs01_0003784 and SASI_Hs02_0034660). Trans-fected cells were incubated for 4 days in 2D cultures followed by harvesting (for IF) or trypsinization and replating into 3D cultures. Crispr/Cas sgRNAs targeting PGD[29] were expressed with lentivirus (Sigma: HSPD0000030958 and HSPD0000030956). Exogenous TXNIP and PGD[29] transgenes were expressed with lentivirus (Sigma ORF: TRCN000476492). For 3D tumoroid assays[29], 2D cultures were trypsinized into single cells and replated in triplicate into 3D formats (1500 cells resuspended in 50 μl Matrigel discs). 3D cultures were incubated with regular feeds for 3–4 weeks prior to harvesting. Glucose uptake from the media was measured over a 2–3-day period in triplicate experiments using an Aviva glucose monitor, and consumption rates calculated[28] as shown in the Source Data File. For PGD inhibitor experiments, cells were treated with dimethyl sulfoxide (DMSO) or 50 μM 6AN (Sigma) for 2–3 days. For GLUT1 inhibitor experiments, cells were treated with DMSO or either 5 μM (samples 2Lv and 2Lg) or 1 μM (all other samples) BAY-876 (Cayman) for 2–3 days. For ACLY inhibitor experiments, samples were treated with either DMSO or 50 μM (samples 38Lv and 38Lg) or 25 μM (all other samples) BMS-303841 (Cayman) for 2 days.

**Mouse procedures**. Athymic nude mice (NU/J, Jackson Laboratories) aged 6–10 weeks were housed under sterile conditions at 21.9 ± 0.8 °C, 45 ± 15% humidity, under 12-h alternating light/dark cycles (7:00 a.m.–7:00 p.m. light; 7:00 p.m.–7:00 a.m. dark). Splenic implantation experiments[66] were conducted in a sterile hood while mice were under isoflurane anesthesia. Briefly, the spleen was released from the abdominal cavity through a small incision. A hemisplenectomy was then performed after the exposed spleen was double clipped inferior to the hilar vessels. Using a 30-gauge needle, trypsinized single-cell suspensions ($1.5 \times 10^6$ cells resuspended in 100 μl phosphate-buffered saline (PBS)) were injected into the resected spleen (still attached to the hilar vessels), followed by a rinse of 100 μl of PBS. After implantation, the hilar vessels were clipped and the hemispleen removed from the animal. The incision was repaired with sutures (peritoneum) and staples (skin). Fourteen weeks after implantation, mice were euthanized and necropsies conducted. The liver and pancreas were photographed and then fixed for ≥24 h in 10% neutral-buffered formalin. After fixation, the organs were thinly sectioned and submitted in their entirety for paraffin embedding, H&E staining of slides, and microscopic histopathologic examination. Animal protocols and procedures were approved by the Vanderbilt Institutional Care and Use Committee.

**Gene expression analyses**. TXNIP transcript levels in patient tissues were examined from previously prepared RNA-seq[33] and SAGE[13] datasets. For RNA-seq, frozen tissue sections were cut from the indicated rapid autopsy patient samples. Tumor-containing portions of each section were identified by histology and regions of interest were macrodissected in triplicate for RNA extraction. RNA was extracted with Trizol (Life Technologies) and purified using Rneasy Plus Mini Kit (Qiagen). After ribosomal depletion, sequencing libraries were prepared using the TruSeq Stranded Total RNA LT kit (Illumina, RS-122-1202). Samples were barcoded and run on a HiSeq 4000 in either 100 bp per 100 bp or 125 bp per 125 bp paired-end run using the HiSeq 3000/4000 SBS kit (Illumina). Output data (FASTQ files) were mapped to the target genome and BAM format files were generated. The expression count matrix from the mapped reads was determined using HTSeq and the generated raw count matrix was normalized using the R/Bioconductor DESeq2 package. The untransformed TXNIP transcript values generated in these data were analyzed either as all samples in conglomerate or after correcting for the observed baseline differences in TXNIP transcript signals detected between patients. The latter (corrected) values were calculated by dividing the TXNIP signals of each individual sample within a given patient by the highest TXNIP signal measured in that patient. For samples with tumor purity estimates, the baseline corrected values were divided by the estimated fraction of tumor DNA present in the tissue section. For analysis of TXNIP transcript expression in previously published SAGE datasets[13], the total number of TXNIP SAGE tags reported for each sample was normalized by dividing them against the total number of SAGE tags generated for the entire sample, as shown in the Source Data File.

For RT-qPCR experiments, RNA was extracted from cultured cells with Trizol and complementary DNA was generated by RT-PCR. Transcripts were quantified in triplicate using real-time qPCR with SYBR green. Primer sequences are listed in Supplementary Table 1. Signals detected in HPDE control cells were used to normalize expression levels across different cell types. For heatmaps, the highest expression level of an individual gene across a given set of experimental conditions was set to 1. Expression levels of the individual gene across the remaining set of experimental conditions were then divided by the largest to obtain fractions of 1. This allowed visualization of all genes together on the heatmap as normalized fold changes ranging from 0 to 1.

**Confocal IF**. Cells were grown on coverslips in 6-well dishes and imaged using a Zeiss LSM 710 confocal microscope with ZEN software (Zeiss, version 2.3). Gain settings were held constant during imaging. Any settings adjustments required for image analysis were applied equally across all images from a given experiment. At least 80% of all cells within a microscopic field were quantified, and two to three (typically three) separate microscopic fields were evaluated for each biological replicate. ImageJ software (version 1.52) was used to quantify signals. TXNIP was quantified by measuring the fluorescence signal intensity within the cytosol. MondoA nuclear-to-cytoplasmic ratios were generated by dividing the fluorescence intensity of MondoA nuclear signal by the fluorescence intensity of cytosolic signal. Percent of cells positive for surface GLUT1 was quantified by first adjusting settings so that sharp signals outlining the cell surface were clearly demarcated from signals within the cytosol. Cells were counted positive if sharp signals outlined the entire outer surface of the cell. Histone acetylation was quantified by first identifying nuclei with Hoechst and measuring the area (size) of each nucleus. The mean gray value of the acetylation IF signal within each nucleus was then measured and the resulting value(s) divided by the area (size) of the nuclei from which each signal was detected.

**Chromatin immunoprecipitation**. For ChIP assays using MondoA antibodies, cells were fixed at 37 °C for 10 min with 1% formaldehyde[28]. After fixation, cells were washed with ice-cold PBS, scraped from the plate, pelleted (4000 r.p.m. for 5 min at 4 °C), and snap frozen in liquid nitrogen for storage at −80 °C. Nuclear extracts were prepared from thawed cell pellets (5 mM PIPES, 85 mM KCl, 0.5% NP-40) and lysed (50 mM Tris, 10 mM EDTA, 1% sodium dodecyl sulfate (SDS)). Lysed nuclei were sonicated for 20 pulses (15–20 s each) using a Branson sonicator instrument. Sonicated chromatin (200 µg) was diluted and incubated overnight with rotation at 4 °C with 10 µg of antibody. Immune complexes were captured with 60 µl Protein A Dynabeads (Life Technologies) for 2 h at 4 °C. Beads were sequentially washed with salt buffer (20 mM Tris pH = 8, 150 mM NaCl, 2 mM EDTA, 1% Triton X-100, 0.1% SDS), detergent buffer (10 mM Tris pH = 8, 250 mM LiCl, 0.5% NP-40, 0.5% deoxycholate, 1 mM EDTA), and TE to remove nonspecific binding, and bound immune complexes eluted from the beads using 1% SDS. Cross-links were reversed with heat (65 °C for 4 h) and protein digested with proteinase K (1–2 h at 45 °C). Input and IP DNA fractions were phenol–chloroform extracted and precipitated with ethanol and glycoblue. DNA was quantified using a Qbit (version 2.0, Life Technologies) and equal amounts of input and IP DNA amplified and quantified by real-time PCR (Roche LightCycler 96) with SYBR Green. For native ChIP (H3K27Ac), purified high molecular weight native chromatin was prepared using a variation of the Workman high salt nuclear extraction method to reduce epitope blocking of acetyl antigens. Briefly, unfixed cells were washed with ice-cold PBS, scraped from plates, and lysed with gentle dounce homogenizer in nuclear extraction buffer (20 mM HEPES, 5 mM MgCl, 250 mM sucrose, 0.2% NP-40, 1 mM dithiothreitol (DTT), and inhibitor cocktail: 5 µM trichostatin A, 5 mM nicotinamide, 10 mM sodium butyrate, Roche Complete EDTA-free protease inhibitors). Nuclei were gently pelleted and washed once with nuclear extraction buffer, pelleted again, and resuspended in solubilization buffer (20 mM HEPES, 325 mM NaCl, 1 mM EDTA, 340 mM sucrose, 0.5 mM DTT, and inhibitor cocktail). Five moles of NaCl was then added dropwise to the solution to bring the final concentration to 650 mM. Nuclei were vigorously broken with 60 strokes in a dounce homogenizer, followed by centrifuging for 20 min at 10,000 × *g*, and supernatants slowly filtered through a 50 K Amicon Ultra spin column (Millipore) for 1 h (2500 × *g* at 4 °C with periodic pipetting up and down in the column) to acquire a concentrated solution of high molecular weight chromatin that is largely stripped of non-nucleosomal protein content. High molecular weight chromatin was diluted in digestion buffer (50 mM Tris, 3 mM CaCl, 340 mM sucrose, inhibitor cocktail) and samples separated into undigested fractions (undigested input) and digested fractions. The latter was then digested into mono-, di-, and tri-nucleosomes with micrococcal nuclease (10 U per 100 µg chromatin) for 5 min at 37 °C, followed by quenching with EDTA/EGTA. NaCl was added dropwise back to each sample (final concentration: 650 mM) to capture any insoluble chromatin fragments, followed by dilution with IP buffer (20 mM Tris, 150 mM NaCl, 5 mM EDTA, 0.01% NP-40, inhibitor cocktail). Digested chromatin was separated into digested input fraction and the IP fraction, and the IP fraction (20 µg) incubated overnight with 10 µg antibody at 4 °C. After antibody capture (Dynabeads, Life Technologies) and washing of the IP fraction, proteins were cleaved with proteinase K and DNA phenol–chloroform extracted with ethanol precipitation. This generated digested input DNA enriched for nucleosomes and IP DNA enriched for acetylated nucleosomes. In parallel, genomic DNA was extracted from the undigested input and sonicated to 150–500 bp. This generated undigested input DNA. DNA concentrations were measured with Qbit (version 2.0, Life Technologies) and equal amounts loaded into 96-well plates for real-time qPCR amplification using SYBR Green (Roche Light-Cycler 96). Enrichments were calculated using the formula: $(2^{\text{Ct undigested input} - \text{Ct IP}} − 2^{\text{Ct undigested input} - \text{Ct digested input}})2$.

**Enzyme assays and metabolite measurements**. PGD enzyme activity assays[29] were conducted on 5–10 µg of cell extracts in triplicate. Briefly, metabolites were filtered out from protein extracts using Amicon Ultra 10 K spin columns (Millipore). Ice-cold filtered extracts were resuspended in ice-cold reaction buffer (50 mM Tris pH = 8, 0.2 mM NADP, 0.4 mM 6PG, 1 mM MgCl₂) and serial

NADPH absorbance measurements (340 nM) were collected from a 96-well plate at 37 °C using a Synergy HTX plate reader (BioTek). NADPH production rates were then calculated from a standard curve as shown in the Source Data File. For 6PG and G6P measurements, metabolites were extracted from cells at −80 °C with 80:20 methanol and water. Metabolites were lyophilized, resuspended, and incubated in triplicate with 0.2 mM NADP and 0.25–1 µg of recombinant protein (PGD or G6PD). Concentrations were calculated by fitting the resulting NADPH production (340 nm absorbance) onto a standard curve and dividing that value by the number of cells counted at the time of metabolite harvest.

**Graphical analysis and statistical tests of significance**. Dot plots, heatmaps, Mann–Whitney *U* tests, ANOVA tests, Fisher's exact tests, and Pearson's correlation coefficients were generated using GraphPad Prism software (version 8.2.1). Bar graphs with dot plot overlays were generated using R software (version 4.0). All other plots and Student's *t* tests were calculated using the Microsoft Excel software.

**Reporting summary**. Further information on research design is available in the Nature Research Reporting Summary linked to this article.

## Data availability

RNA-seq data analyzed in this study are deposited in the European Genome-Phenome Archive under the accession code EGAS00001003974. Published gene sets analyzed in this study are available from previous papers[13,28,33] and can be found at the following URLs: https://www.ncbi.nlm.nih.gov/geo/query/acc.cgi?acc=GSE63126 and https://science.sciencemag.org/content/suppl/2008/09/04/1164368.DC1?_ga=2.19445630.711824025.1593795472-1581683601.1591377453. All the other data supporting the findings of this study are available within the article and its Supplementary Information files and from the corresponding author upon reasonable request. A reporting summary for this article is available as a Supplementary Information file. No new computer code was required to analyze this data or derive the conclusions presented in this manuscript. Source data are provided with this paper.

## Code availability

No new computer code was required to analyze this data or derive the conclusions presented in this manuscript.

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

## Acknowledgements

We thank J. Rathmell and R. Coffey for helpful discussions, and members of the Vanderbilt TPSR for histopathology technical assistance. This work was supported by National Institutes of Health Grants R01 CA222594 (O.G.M.), R01 CA174761 (K.E.W.), and F31 CA217070-01 (S.L.C.), the Vanderbilt GI SPORE 5P50 CA95103 (O.G.M.), the Vanderbilt Digestive Diseases Research Center 5P30 DK058404-17 (O.G.M.), and the American Cancer Society IRG-58-009-54 (O.G.M.).

## Author contributions

M.E.B. and O.G.M. performed cell culture experiments. M.E.B., Y.Z., V.L.W., and O.G.M. performed mouse experiments. O.G.M. conceived the work, oversaw experiments and data analysis, and wrote the manuscript. R.S. performed enzyme assays and metabolite measurements. A.H. generated RNA-seq data. A.H. and C.I.-D. advised on analysis of RNA-seq. A.E.W. and A.V.T. performed qPCRs, plotted data, and conducted statistical analyses. S.L.C. and K.E.W. assisted with design of glucose uptake experiments and performed YSI quality control confirmation of Aviva results. O.G.M. performed histopathologic analysis and immunohistochemical scoring. O.G.M. and M.E.B. assembled the figures. All authors agreed to the final version of the manuscript.

## Competing interests

The authors declare no competing interests.
