## [Peer Review File · Nature Communications]

Reviewers' Comments:

Reviewer #1:

Remarks to the Author:

This study by Becharad et al builds on the authors' previous findings that cells derived from distant metastases of pancreatic ductal adenocarcinoma (PDAC) are specifically dependent on the oxidative pentose phosphate pathway (ox-PPP) enzyme, phosphogluconate dehydrogenase (PGD), for growth in vitro. This work suggested that a distinct metabolic program supports metastatic spread and is linked to extensive changes in chromatin regulation. Since multiple studies show that primary PDAC and metastasis share the same genetic profiles, the exploration of the non-genetic basis of metastatic outgrowth and dependency is an important area of study and the role of altered metabolic pathways in the survival of distant metastatic PDACs is a new paradigm in metastasis research. In the present study the authors have examined the impact of PDG on the regulation of glucose uptake. They find that the presence of high levels of PDG in cells derived from distant metastasis generates a positive feedback pathway to maintain high glucose uptake. They report that thioredoxin-interacting protein (TXNIP), an inhibitor of cellular glucose uptake, is reduced in these cells compared to cells from peritoneal metastasis, and that PGD knockdown increases TXNIP levels and suppresses glucose uptake. They relate these effects to the inhibition of the MondoA transcription factor by accumulation of the PDG substrate, 6PG. They also show that TXNIP overexpression specifically reduces the growth of 3D organoids from distant metastasis. Finally, the authors show in patient tumor samples, that TXNIP staining is reduced distant metastases compared to peritoneal metastases.

These studies are novel and of interest and the data showing the relationship between PDG, TXNIP and glucose uptake are clear-cut. The primary concern is that limitations in the model systems restrict the conclusions that can be drawn. The study is primarily framed around the difference between peritoneal and distant metastasis. Moreover, most experiments compare a single pair of peritoneal vs distant metastasis models. It is not clear whether the differences noted relate to features particular to the peritoneal metastasis used as a reference rather than properties acquired as primary tumors undergo distant metastatic spread. Also, the generality of the observations is not established. Finally, the cell lines used in this manuscript do not form tumors in xenograft assays, and so their relevance to the in vivo situation is uncertain. While it is understood that these models are difficult to obtain, some additional support would be needed to confirm the conclusions of the paper.

- 1) Figure 1B should include additional primary tumor-derived lines in the comparison. The present analysis includes long-established cell lines from repositories and a single peritoneal met sample as references. A more relevant and complete comparison would utilize matched early passage primary and mets (preferably) or a number of early passage primary PDA lines.
- 2) The functional studies are based on a single pair of cell lines (38Per vs 38Lg). The phenotypes should be tested in multiple cell lines.
- 3) Please include immunoblots for TXNIP (Fig 1D) and the histone marks (Fig 3a).
- 4) The rationale for excluding primary tumors in the IHC analysis in Figure 4 is not clear. As noted above, it is not certain whether TXNIP is decreased in metastasis compared to primary PDAC or whether TXNIP levels are specifically elevated in peritoneal metastases.
- 5) The regulation of nucl-cyto translocation of MondoA by 6PG is an important observation (Extended data 3). The authors should clarify how these metabolites are treated in this experiment. Since phosphorylated compounds like 6PG and G6P are poorly cell-permeable, it is important to show the intracellular concentration of the 3PG after exogenous addition.
- 6) Are steady-state 6PG levels reduced in the PGD-high cells? Also, in
- 7) The impact of the study would be enhanced by some insight into how inhibition of PGD regulates mondoA-TXNIP axis?
- 8) In Figure 4 the authors used siRNA to knockdown TXNIP and performed 3D growth assays over 7-21 days. Since siRNA causes transient knockdown the authors may want to consider additional controls.

Reviewer #2:

Remarks to the Author:

This is an interesting and well done manuscript that describes a role for altered metabolism in distant pancreatic cancer metastatic lesions. This manuscript follows others from the same group and should be of high general interest. The data is convincing and clearly shows that glucose uptake is upregulated in distant lesions that are dependent on PGD and that high PDG levels result in suppression of TXNIP. This pathway dissection is followed up by a reasonable set of experiments to validate the biological significance of their working model. Particularly interesting is the finding that in PDG tumors the levels of H3K27ac are upregulated, perhaps suggesting a broad rewiring of gene expression in this subgroup of lesions. On the downside, the manuscript is largely descriptive and lacks mechanistic depth. This is a major concern.

1. The proposal is that TXNIP loss leads to an increase in glucose uptake, which is consistent with a body of literature and clearly shown here. They propose that the increase in glucose uptake leads to an increase in Acetyl-CoA resulting in an increase levels of histone acetylation. which is not explicitly tested here. This is important given TXNIP's many roles that

seem to be unrelated to functions in glucose uptake and/or metabolism. For example, TXNIP loss in MEFs leads to activation of PI3-Kinase, which could easily account for their observations.

2. The staining for H3K27Ac is convincing as is the ChIP profiling for H3K27Ac at CDH2 shown in Figure 3b. At least a few more TXNIP regulated targets shown in figure 3b, should be evaluated. It may be just as easy to perform a ChIP-seq experiment for H3K27ac.

3. PGD loss increases TXNIP expression and the authors' have some data that induction is MondoA-dependent. They should test this directly and determine whether TXNIP induction depends on MondoA as they data as it now stands is only correlative. It would also be valuable to determine if PGD loss leads to an increase in G6P as this is the primary regulator of MondoA transcriptional activity. Likewise measuring G6P levels in PGDhigh lesions would also be informative. Determining how G6P regulates MondoA activity is a big ask and is beyond the scope of this manuscript.

4. This is more of a minor concern. The author's present their data as consistent with both positive and negative feedback loops. Clearly this is complicated, but the description was cumbersome in my opinion and they don't really test the regulatory loops rigorously. It may be more straight forward just to describe the metabolic wiring of cells in the PDGhigh and the PDGlow states.

Reviewer #3:

Remarks to the Author:

Bechard et al study distant vs peritoneal metastasis in PDAC. They find that distant metastasis with high PGD (identified in a previous paper) have low expression of TXNIP. Reduced expression of TXNIP is known to lead to increased GLUT1 expression and therefore glucose uptake. Accordingly, they find that PGD regulated TXNIP expression impacts on glucose uptake. In line, exogenous TXNIP decreased 3D growth of distant metastasis but not peritoneal metastasis (while TXNIP silencing impair peritoneal metastasis) and TXNIP was less expressed in distant metastasis from patients. The main finding of this paper is that PGD regulates (by an unknown mechanism) TXNIP expression and that this decreased expression is important for the previously described change in histone acetylation.

Major comments:

- 1) The focus on a switch from a negative to a positive feedback loop is confusing and not supported by data. In my opinion, the data simply suggest that PGD is a negative regulator of TXNIP.
- 2) Accordingly, it is required that title and abstract reflect the actual data presented in the manuscript rather than a speculative interpretation of the data.
- 3) The authors mention several times the context of evolution and non-genetic driver. These terms should be removed from the manuscript or experiments should be provided to support the claims. For example, is PGD overexpression or TXNIP deletion sufficient to drive a non-metastatic PDAC cell to lung or liver?
- 4) Evidence should be provided that altering PGD or TXNIP prevents in vivo metastasis formation.

Reviewers' comments:

Reviewer #1 (Remarks to the Author):

This study by Bechard et al builds on the authors' previous findings that cells derived from distant metastases of pancreatic ductal adenocarcinoma (PDAC) are specifically dependent on the oxidative pentose phosphate pathway (ox-PPP) enzyme, phosphogluconate dehydrogenase (PGD), for growth in vitro. This work suggested that a distinct metabolic program supports metastatic spread and is linked to extensive changes in chromatin regulation. Since multiple studies show that primary PDAC and metastasis share the same genetic profiles, the exploration of the non-genetic basis of metastatic outgrowth and dependency is an important area of study and the role of altered metabolic pathways in the survival of distant metastatic PDACs is a new paradigm in metastasis research. In the present study the authors have examined the impact of PGD on the regulation of glucose uptake. They find that the presence of high levels of PGD in cells derived from distant metastasis generates a positive feedback pathway to maintain high glucose uptake. They report that thioredoxin-interacting protein (TXNIP), an inhibitor of cellular glucose uptake, is reduced in these cells compared to cells from peritoneal metastasis, and that PGD knockdown increases TXNIP levels and suppresses glucose uptake. They relate these effects to the inhibition of the MondoA transcription factor by accumulation of the PGD substrate, 6PG. They also show that TXNIP overexpression specifically reduces the growth of 3D organoids from distant metastasis. Finally, the authors show in patient tumor samples, that TXNIP staining is reduced distant metastases compared to peritoneal metastases.

These studies are novel and of interest and the data showing the relationship between PGD, TXNIP and glucose uptake are clear-cut. The primary concern is that limitations in the model systems restrict the conclusions that can be drawn. The study is primarily framed around the difference between peritoneal and distant metastasis. Moreover, most experiments compare a single pair of peritoneal vs distant metastasis models. It is not clear whether the differences noted relate to features particular to the peritoneal metastasis used as a reference rather than properties acquired as primary tumors undergo distant metastatic spread. Also, the generality of the observations is not established. Finally, the cell lines used in this manuscript do not form tumors in xenograft assays, and so their relevance to the in vivo situation is uncertain. While it is understood that these models are difficult to obtain, some additional support would be needed to confirm the conclusions of the paper.

We thank the reviewer for the positive evaluation and helpful critiques. We have attempted to comprehensively address all areas of concern as detailed point-by-point below, including addition of primary tumor samples, establishing generality and recurrence of the phenotypes, and new *in vivo* experimental metastasis data.

1) Figure 1B should include additional primary tumor-derived lines in the comparison. The present analysis includes long-established cell lines from repositories and a single peritoneal met sample as references. A more relevant and complete comparison would utilize matched early passage primary and mets (preferably) or a number of early passage primary PDA lines.

Primary tumor samples were originally omitted because the regional/subclonal heterogeneity that is well-known to exist in PDAC primaries complicates drawing conclusions in the absence of large numbers of patients or multi-regional sequencing (MRS) within individual patients. We also note that prospective isolation of new cell lines

from matched primary tumor and metastases in the same patient(s) (by additional rapid autopsies for example) is far beyond the scope and time allotted for this study, especially since culture take rates from primary patient PDACs are relatively low in our experience (compared to metastases). We therefore undertook several alternative approaches to address these important issues.

First, our MSKCC collaborators kindly provided TXNIP expression data from (at the time) their unpublished RNA-seq experiments conducted on 127 matched primary tumor (n=95) and liver metastasis (n=32) tissue samples. These data (page 5, lines 107-129) conclusively addressed the question of TXNIP status in matched primary tumors and liver metastases from the same patient(s) while also capturing primary tumor heterogeneity since MRS was applied to each primary. Irrespective of how the data was analyzed, TXNIP was higher overall in the aggregated primary tumor samples (Fig. 1e-f, Supplementary Fig. 2). A subset of instructive phylogenetic trees was also available that very clearly illustrated these conclusions (Fig. 1g, Supplementary Fig. 2e-f), and TXNIP IHC studies confirmed that protein expression matched the RNA-seq for the patient highlighted in Fig. 1g, as formalin fixed sections were serendipitously still available from the same tissue as the frozen sections used for RNA-seq.

As an independent validation of the RNA-seq, we also examined a different dataset (Jones et al. Science 2008, page 4-5 lines 104-106). These data were ideal since they included a subset of the same rapid autopsy metastases from our manuscript along with 23 additional primary tumor samples. Even though these were assayed using a different technique (SAGE) performed on bulk (rather than multi-regional) primary tumors, TXNIP was again significantly higher in the primaries (Supp Fig. 2a).

To ease concerns that data presented in Figs. 1b and 1d did not include primary tumor samples to complement the peritoneal lines, we identified a separate set of PGD-independent primary tumor lines (verified in Supp Fig. 1b). These are now included in Fig. 1b and 1d, with similar findings as the peritoneal controls.

We note that one of the PGD-dependent lines from our original manuscript was originally isolated from a primary tumor subclone (13Pr) that shares similar phenotypic properties with a distant metastasis from the same patient (13Lg). This is not unexpected given the regional heterogeneity discussed above, especially if PGD-dependence and metastatic potential are acquired during subclonal evolution in the primary. Unlike the other lines derived from pre-established liver/lung mets (including 13Lg), we have found that 13Pr can form metastatic tumors in nude mice. 13Pr is now highlighted in the manuscript (page 11-13, lines 274-311), including (re)-engineering the PGD sgRNAs and TXNIP transgenes. Those interventions replicated all previously shown phenotypes and provided convincing *in vivo* evidence that inactivating PGD or restoring TXNIP prevented experimental metastasis in athymic mice (Fig. 7).

2) The functional studies are based on a single pair of cell lines (38Per vs 38Lg). The phenotypes should be tested in multiple cell lines.

Beyond the expression data presented in Fig. 1, the various phenotypes reported in the manuscript are now tested for generality/recurrence across the larger cohort of rapid autopsy lines with a large number of new experiments (Fig. 2f, Fig. 4f, Fig. 5d, Fig. 7b-e, Supp Fig. 1a-b, Supp Fig. 6, Supp Fig. 8, Supp Fig. 10).

3) Please include immunoblots for TXNIP and the histone marks (Fig 3a).

Immunoblots for the requested experiments are now included in Supp Fig. 7b and Supp Fig. 9a, which match results of the IF panels presented in the main figures (Fig. 6b). As an aside, our lab has transitioned from previously conducting mostly immunoblots to now conducting mostly confocal IF since (in our hands) the IF is less labor intensive, more cost effective, highly reproducible, better suited for quantification of biological replicates, and provides more information regarding the subcellular location(s).

4) The rationale for excluding primary tumors in the IHC analysis in Figure 4 is not clear. As noted above, it is not certain whether TXNIP is decreased in metastasis compared to primary PDAC or whether TXNIP levels are specifically elevated in peritoneal metastases.

Please refer to our response to comment #1. Primary tumor MRS with matched liver metastases from the same individual patients were an ideal data platform to address this question and the results were clear. The IHC findings for liver vs. peritoneal mets are still included in the manuscript, as only a few matched peritoneal metastases were available from the RNA-seq study (Fig. 1g, with IHC of matching primary). While primary tumors in aggregate express higher TXNIP than matched liver mets, the reviewer may be correct that TXNIP is up-regulated even further in peritoneal metastases. We are currently preparing a follow-up manuscript to address the role of TXNIP in primary and peritoneal PDACs, as briefly alluded to in the discussion (page 13-14, lines 337-338).

5) The regulation of nucl-cyto translocation of MondoA by 6PG is an important observation (Extended data 3). The authors should clarify how these metabolites are treated in this experiment. Since phosphorylated compounds like 6PG and G6P are poorly cell-permeable, it is important to show the intracellular concentration of the 6PG after exogenous addition.

Although phosphorylated sugars are impermeable to exit cells once they are inside, we have observed that these metabolites are in fact taken up into cells when added to the media exogenously. We apologize for omitting the important control measurements demonstrating intracellular uptake of these metabolites in the original manuscript. They are now included for both 6PG and G6P with the MondoA translocation data in Fig. 4g.

6) Are steady-state 6PG levels reduced in the PGD-high cells?

Yes, steady state 6PG is recurrently depleted in the rapid autopsy PGD^{high} cells by high resolution LC-MS metabolomics (McDonald et al. Nature Genetics 2017). In fact, it was this observation that initially inspired us to begin investigating PGD in the first place. This is now more clearly introduced in the background text (page 4 lines 81-89). To provide further support, new data is presented in Supp. Fig. 1a demonstrating that 6PG is also the most depleted metabolite in LC-MS data collected on a separate subset of PGD-dependent cell lines from the cancer cell line encyclopedia.

7) The impact of the study would be enhanced by some insight into how inhibition of PGD regulates mondoA-TXNIP axis?

We have now devoted an entire section and main Figure with several pieces of new data (Fig. 4), along with additional new Supplementary data (Supp Fig. 8) to address this point (pages 8-9, lines 193-233). We have also provided additional insights into how we think this interesting mechanism might work in the text (page 9, lines 227-233).

8) In Figure 4 the authors used siRNA to knockdown TXNIP and performed 3D growth assays over 7-21 days. Since siRNA causes transient knockdown the authors may want to consider additional controls.

As the reviewer points out, siRNAs are transient knockdowns. Nevertheless, in our experience the effects on tumorigenic growth are durable for several weeks when cells are pre-treated with siRNAs for 4 days in 2D, followed by trypsinization and (re)-plating into 3D formats (for example, Figs. 1, 3, 4, and Supp Figs 3 and 4 from Bechard et al. Oncogene, Fig. 7 from McDonald et al NG). We have recently obtained identical results as previously shown for the siRNAs using both stable shRNA knockdown of TXNIP and Crispr/Cas deletion of TXNIP. We are happy to share these data with the reviewer if he/she feels it necessary. It is no longer included in this manuscript because we have moved all TXNIP loss-of-function experiments into a follow-up manuscript focused on how TXNIP supports primary tumor growth and peritoneal metastasis.

Reviewer #2 (Remarks to the Author):

This is an interesting and well done manuscript that describes a role for altered metabolism in distant pancreatic cancer metastatic lesions. This manuscript follows others from the same group and should be of high general interest. The data is convincing and clearly shows that glucose uptake is upregulated in distant lesions that are dependent on PGD and that high PDG levels result in suppression of TXNIP. This pathway dissection is followed up by a reasonable set of experiments to validate the biological significance of their working model. Particularly interesting is the finding that in PDG tumors the levels of H3K27ac are upregulated, perhaps suggesting a broad rewiring of gene expression in this subgroup of lesions. On the downside, the manuscript is largely descriptive and lacks mechanistic depth. This is a major concern.

We thank the reviewer for the positive manuscript review and thoughtful critiques for improvement. We have attempted to address all concerns as outlined point-by-point below, including a large amount of new experiments designed to more rigorously test the various mechanistic aspects of this work.

1. The proposal is that TXNIP loss leads to an increase in glucose uptake, which is consistent with a body of literature and clearly shown here. They propose that the increase in glucose uptake leads to an increase in Acetyl-CoA resulting in an increase levels of histone acetylation, which is not explicitly tested here. This is important given TXNIP's many roles that seem to be unrelated to functions in glucose uptake and/or metabolism. For example, TXNIP loss in MEFs leads to activation of PI3-Kinase, which could easily account for their observations.

As the reviewer astutely notes, in the original manuscript it was implied that loss of TXNIP facilitated a glucose-fueled PGD-GLUT1-ACLY hyperacetylation pathway. However, this mechanism was not directly tested with experiments. An entire new section with large numbers of experiments is now presented in the revision indicating that glucose, PGD, GLUT1, and ACLY are each required to maintain hyperacetylation across the full cohort of PGD^{high} samples (Fig. 5, pages 10-11 lines 234-273). Moreover, new data shows that TXNIP and MondoA knockdowns each strongly rescued loss of acetylation during PGD inactivation (Fig. 6a).

2. The staining for H3K27Ac is convincing as is the ChIP profiling for H3K27Ac at CDH2 shown in Figure 3b. At least a few more TXNIP regulated targets shown in figure 3b, should be evaluated.

New ChIP experiments are now included in Fig. 6d-f of the revised manuscript to complement CDH2, including ODC1 (one of the most highly expressed metastatic genes) and FOXA1 (an important pro-metastatic transcription factor).

3. PGD loss increases TXNIP expression and the authors' have some data that induction is MondoA-dependent. They should test this directly and determine whether TXNIP induction depends on MondoA as they data as it now stands is only correlative. It would also be valuable to determine if PGD loss leads to an increase in G6P as this is the primary regulator of MondoA transcriptional activity. Likewise measuring G6P levels in PGD^{high} lesions would also be informative. Determining how G6P regulates MondoA activity is a big ask and is beyond the scope of this manuscript.

To address MondoA dependence, new data is presented showing that MondoA knockdown rescues the effects of PGD inactivation on TXNIP, GLUT1, (Fig. 4e) and histone acetylation (Fig. 6a). These experiments confirm that TXNIP induction and the resulting phenotypes are indeed MondoA-dependent. We note that these results mirror those obtained during similar rescues by TXNIP knockdown (Fig. 2e, Fig. 6a).

Regarding the reviewer's astute questions regarding G6P and 6PG, our previous LC-MS experiments identified 6PG as the sole glucose-derived metabolite that is depleted in PGD^{high} cells (McDonald et al. NG 2017). Subsequent experiments have also indicated that G6P does not accumulate in any of our PGD^{high} cells during PGD inhibition. This large amount of negative G6P data is not presented in the revision for space, although we can include it if the reviewer feels strongly about it. 6PG on the other hand strongly accumulates during PGD inhibition as MondoA traffics back to the nucleus (Supp Fig. 8 of the current revised manuscript, see also McDonald et al. Nature Genetics 2017).

We are aware of previous studies that propose G6P as a regulator of MondoA trafficking in other cell systems. However, in the PGD^{high} setting examined here it appears that 6PG supersedes G6P for this purpose, as demonstrated by directly evaluating the effects of 6PG and G6P in Fig. 4g (including confirmation of intracellular metabolite uptake). Results in Fig. 4 are further supported with new data presented in Supp Fig. 8. Please refer to our responses to reviewer 1 (comments 5-7) for additional points related to these general topics. We appreciate that the reviewer does not expect us to solve the longstanding (and peripheral) question of how transcription factors physically sense and respond to metabolites (now pointed out in the text on page 9 lines 228-233).

4. This is more of a minor concern. The author's present their data as consistent with both positive and negative feedback loops. Clearly this is complicated, but the description was cumbersome in my opinion and they don't really test the regulatory loops rigorously. It may be more straight forward just to describe the metabolic wiring of cells in the PDG^{high} and the PDG^{low} states.

We thank the reviewer for the very useful suggestion of PGD^{high} and PGD^{low} activation status, which are incorporated throughout the revised manuscript text and figures. Feedback loop descriptors are also largely removed from the revised manuscript except

where necessary as background information. Despite the revised text, we still agreed that it was important to more rigorously test the feedback loops. To this end, we conducted large numbers of new experiments that enriched and greatly expanded the data presented in the revised manuscript, as briefly summarized below.

To further test PGD-driven suppression of a MondoA-TXNIP-GLUT1 negative feedback loop, we provide new supporting evidence that TXNIP is recurrently suppressed across larger numbers of patient metastases *in vivo* (Fig. 1), that PGD regulates MondoA trafficking recurrently across the larger cohort of rapid autopsy patients (Supp. Fig. 8, Fig. 4f), and that MondoA directly mediates the previously observed changes in TXNIP, GLUT1, and histone acetylation that occur during PGD inactivation as conclusively shown by rescue experiments (Fig. 4e, Fig. 6a).

To further test activation of a PGD-GLUT1-Glucose positive feedback loop, we provide additional data showing: (i) PGD regulates glucose uptake (Fig. 2c, Fig. 2f, Supp Fig. 10b) and GLUT1 surface retention (Supp Fig. 6b, Fig. 2d-e, Fig. 7b), (ii) GLUT1 regulates glucose consumption (Supp Fig. 6a) and PGD activity (Supp Fig. 7c), and (iii) glucose and TXNIP (which regulates GLUT1/glucose) both regulate PGD activity (Supp Fig. 7d, Fig. 3c). By extension, new experiments further demonstrate that PGD, GLUT1, and glucose are each (in concert with ACLY) required to maintain histone hyperacetylation (Fig. 5), while TXNIP opposes this phenotype (Fig. 6, Fig. 7c, Supp Fig. 9a).

Reviewer #3 (Remarks to the Author):

Bechard et al study distant vs peritoneal metastasis in PDAC. They find that distant metastasis with high PGD (identified in a previous paper) have low expression of TXNIP. Reduced expression of TXNIP is known to lead to increased GLUT1 expression and therefore glucose uptake. Accordingly, they find that PGD regulated TXNIP expression impacts on glucose uptake. In line, exogenous TXNIP decreased 3D growth of distant metastasis but not peritoneal metastasis (while TXNIP silencing impair peritoneal metastasis) and TXNIP was less expressed in distant metastasis from patients. The main finding of this paper is that PGD regulates (by an unknown mechanism) TXNIP expression and that this decreased expression is important for the previously described change in histone acetylation.

We thank the reviewer for critically evaluating the manuscript and providing helpful comments. We have addressed the critiques with new data and revised text as outlined point-by-point below, including toning down the language to match the data and providing evidence that altering PGD or TXNIP prevents experimental metastasis in vivo.

Major comments:

1) The focus on a switch from a negative to a positive feedback loop is confusing and not supported by data. In my opinion, the data simply suggest that PGD is a negative regulator of TXNIP.

The feedback exchange terminology has been removed from the revised manuscript, except in the final paragraph of the discussion (page 14, lines 342-354). That lone paragraph remains for two reasons. First, we have included a large amount of new experiments throughout the revision that more rigorously interrogate both the negative

feedback (MondoA-TXNIP-GLUT1) and positive feedback (PGD-GLUT1-Glucose) components (see response to reviewer #2 comment 4). Second, in our opinion a positive feedback loop that suppresses its negative feedback opposition is an important yet difficult concept that requires a capture phrase (“feedback exchange” or similar) to simplify discourse, especially if similar processes involving different nutrients or metabolic enzymes are observed by other investigators. We are open to alternative terminology suggestions if a more effective capture phrase is proposed.

2) Accordingly, it is required that title and abstract reflect the actual data presented in the manuscript rather than a speculative interpretation of the data.

The title and abstract have been revised to focus more squarely on the data.

3) The authors mention several times the context of evolution and non-genetic driver. These terms should be removed from the manuscript or experiments should be provided to support the claims. For example, is PGD overexpression or TXNIP deletion sufficient to drive a non-metastatic PDAC cell to lung or liver?

The overstated driver terminology has been replaced in the revision with “dependency” or “adaptation” when referencing the various PGD-dependent metastatic properties.

The experiment to express PGD in a non-metastatic cell is a thoughtful suggestion. It would be included if PGD-dependence and metastasis were dictated by PGD expression. However, PGD is already expressed in virtually all PDACs (irrespective of site), and PGD is not typically over-expressed (or mutated or amplified) in metastases. Rather, it is the catalytic activity that is high (Bechard et al. Oncogene). We have now denoted this as “PGD^{high}” (suggested by reviewer 2) in the revision, with text clarifications at various points (page 2 lines 30-32, page 3 lines 65-67, page 4 lines 81-89, page 13 lines 313-319).

Because the metastatic cells have acquired new shunting mechanisms (Bechard et al. 2018 and other unpublished work) that route the avid glucose uptake (described in the current work) to PGD, all these ancillary supports will need to be fully reconstituted in order to convert a non-metastatic cell to PGD^{high}. We hope the reviewer understands that achieving this is a tall order that will require considerable time and labor that far exceeds the scope of the current work.

When we knocked down or deleted TXNIP in PGD^{low} PDACs, it completely abolished the ability of those cells to form tumors (with no effect on PGD^{high} cells, which do not express it). Because this intriguing result is the mirror image of what we report in this manuscript for PGD^{high} PDACs, we are preparing what we believe is an equally interesting follow-up manuscript addressing the role(s) of TXNIP for primary tumor growth and peritoneal metastasis, as briefly alluded to in the discussion (pages 13-14 lines 337-338).

4) Evidence should be provided that altering PGD or TXNIP prevents in vivo metastasis formation.

Evidence for both is now presented in Fig. 7 (pages 11-13, lines 274-311).

Reviewers' Comments:

Reviewer #3:

Remarks to the Author:

The authors sufficiently addressed my concerns.

Reviewer #4:

Remarks to the Author:

The authors have answered most of the questions raised during the first course of reviewing. The manuscript has improved. No more comments.

REVIEWERS' COMMENTS:

Reviewer#3:

The authors sufficiently addressed my concerns.

We thank the reviewer for critiquing the manuscript and accepting our revisions.

Reviewer#4: (Reviewer#2 replacement)

The authors have answered most of the questions raised during the first course of reviewing.

The manuscript has improved. No more comments.

We thank the reviewer for critiquing the manuscript and accepting our revisions.